# Inter-laboratory comparison of $\delta^{13}C$ and $\delta D$ measurements of atmospheric $CH_4$ for combined use of datasets from different laboratories

Taku Umezawa[1,2], Carl A. M. Brenninkmeijer[1], Thomas Röckmann[3], Carina van der Veen[3], Stanley C. Tyler[4,5], Ryo Fujita[6], Shinji Morimoto[6,7], Shuji Aoki[6], Todd Sowers[8], Jochen Schmitt[9], Michael Bock[9], Jonas Beck[9], Hubertus Fischer[9], Sylvia E. Michel[10], Bruce H. Vaughn[10], John B. Miller[10], James W. C. White[10], Gordon Brailsford[11], Hinrich Schaefer[11], Peter Sperlich[11], Willi A. Brand[12], Michael Rothe[12], Thomas Blunier[13], David Lowry[14], Rebecca E. Fisher[14], Euan G. Nisbet[14], Andrew L. Rice[15], Peter Bergamaschi[16], Cordelia Veidt[17] and Ingeborg Levin[17]

[1]Max Planck Institute for Chemistry, 55128 Mainz, Germany
[2]National Institute for Environmental Studies, Tsukuba 305-8506, Japan
[3]Institute for Marine and Atmospheric research Utrecht, Utrecht University, Utrecht, The Netherlands
[4]Earth System Science Department, University of California, Irvine, USA
[5]Chemistry Department, Norco College, Norco, CA 92860, USA
[6]Center for Atmospheric and Oceanic Studies, Graduate School of Science, Tohoku University, Sendai, Japan
[7]National Institute of Polar Research, Tokyo, Japan
[8]Pennsylvania State University, University Park, Pennsylvania 16802, USA
[9]Climate and Environmental Physics, Physics Institute and Oeschger Center for Climate Change Research, University of Bern, 3012 Bern, Switzerland
[10]Institute of Arctic and Alpine Research, University of Colorado Boulder, Boulder, Colorado, USA
[11]National Institute of Water and Atmospheric Research, Wellington 6021, New Zealand
[12]Max Planck Institute for Biogeochemistry, 07745 Jena, Germany
[13]Center for Ice and Climate, University of Copenhagen, Copenhagen, Denmark
[14]Department of Earth Sciences, Royal Holloway, University of London, Egham, UK
[15]Department of Physics, Portland State University, Portland, OR 97207, USA
[16]European Commission Joint Research Centre, Ispra (Va), Italy
[17]Institute for Environmental Physics, Heidelberg University, 69120 Heidelberg, Germany

*Correspondence to*: Taku Umezawa (umezawa.taku@nies.go.jp)

**Abstract.** We report results from a worldwide inter-laboratory comparison of samples among laboratories that measure (or measured in the past) stable carbon and hydrogen isotope ratios of atmospheric $CH_4$ ($\delta^{13}C$-$CH_4$ and $\delta D$-$CH_4$). The offsets among the laboratories are larger than the

measurement reproducibility of individual laboratories. To disentangle plausible measurement offsets, we evaluated and critically assessed a large number of intercomparison results, some of which have been documented previously in the literature. The results indicate significant offsets of $\delta^{13}$C-CH$_4$ and $\delta$D-CH$_4$ measurements among datasets reported from different laboratories; the differences among

laboratories at modern atmospheric CH$_4$ level spread over ranges of 0.5 ‰ for $\delta^{13}$C-CH$_4$ and 13 ‰ for $\delta$D-CH$_4$. The intercomparison results summarized in this study may be of help for future attempts to harmonize $\delta^{13}$C-CH$_4$ and $\delta$D-CH$_4$ datasets from different laboratories in order to jointly incorporate them into modelling studies. However, establishing such a merged dataset, which includes $\delta^{13}$C-CH$_4$ and $\delta$D-CH$_4$ data from multiple laboratories with desirable compatibility, is still challenging due to

differences among laboratories in instrument settings, correction methods, traceability to reference materials and long-term data management. Further efforts are needed to identify causes of the inter-laboratory measurement offsets and to decrease those towards the best use of available $\delta^{13}$C-CH$_4$ and $\delta$D-CH$_4$ datasets.

## 1 Introduction

Methane (CH$_4$) is an important anthropogenic and natural greenhouse gas, and it also has a large role in atmospheric chemistry through its reaction with the hydroxyl radical. Since individual CH$_4$ source types have characteristic isotope signatures and loss processes are associated with specific kinetic isotope effects, carbon and hydrogen isotope ratios of CH$_4$ ($\delta^{13}$C-CH$_4$ and $\delta$D-CH$_4$) have been useful to constrain the global CH$_4$ budget. Dictated by global mass balance, the average isotopic composition of

CH$_4$ in the atmosphere ($\delta^{13}$C-CH$_4$ or $\delta$D-CH$_4$) equals the flux-weighted isotopic composition of the sources, corrected for the total kinetic isotope effects of removal processes (e.g. Stevens and Rust, 1982; Cicerone and Oremland, 1988; Quay et al., 1991, 1999; Miller et al., 2002; Turner et al., 2017; Rigby et al., 2017). It has been pointed out that assignment of representative isotopic signatures of various CH$_4$ sources remains uncertain due to their large spatial and temporal variability across the globe (e.g.

Sherwood et al., 2017), which could result in large uncertainties of isotope-based estimates of the global CH$_4$ budget (Schwietzke et al., 2016). Nonetheless, the value of isotope measurements was amply demonstrated by recent studies which suggested shifts in the global CH$_4$ source over the last decades

(Schaefer et al., 2016; Rice et al., 2016; Nisbet et al., 2016; Schwietzke et al., 2016); without isotopic analyses such conclusions would have been difficult to achieve. The isotopic ratios are commonly reported using the delta notation:

$$\delta = \frac{R_{sample}}{R_{standard}} - 1 \tag{1}$$

where $R$ represents the atomic ratio of the less abundant over the most abundant isotope in the sample and the standard, respectively. Conventionally, measured values are reported relative to the international isotope scales VPDB for $\delta^{13}$C-CH$_4$ and VSMOW for $\delta$D-CH$_4$ in per mil.

Given that the atmospheric lifetime of CH$_4$ is about a decade, its variation in background air is relatively small. For that reason, its mole fraction and isotopic measurements have to be made with high
precision and accuracy. For $\delta^{13}$C-CH$_4$ and $\delta$D-CH$_4$, researchers have achieved measurement reproducibility of < 0.1 ‰ for $\delta^{13}$C-CH$_4$ and < 2 ‰ for $\delta$D-CH$_4$. Incorporating $\delta^{13}$C-CH$_4$ and $\delta$D-CH$_4$ datasets in chemistry transport models is useful to quantitatively separate different CH$_4$ source categories and such attempts have been contributed to reduction of uncertainties in the global CH$_4$ budget (e.g. Fung et al., 1991; Hein et al., 1997; Mikaloff Fletcher et al., 2004a, 2004b; Monteil et al.,
2011; Kirschke et al., 2013; Ghosh et al., 2015; Rice et al., 2016; Schaefer et al., 2016; Schwietzke et al., 2016; Röckmann et al., 2016; Turner et al., 2017; Rigby et al., 2017). However, although an increasing number of $\delta^{13}$C-CH$_4$ and $\delta$D-CH$_4$ data has been reported over the last decades, significant measurement offsets among laboratories have been found for both $\delta^{13}$C-CH$_4$ (e.g. Levin et al., 2012) and $\delta$D-CH$_4$ (Bock et al., 2014). It is clear that both traceability to the standard scales and inter-
laboratory comparisons (intercomparisons) are indispensable for combined use of $\delta^{13}$C-CH$_4$ and $\delta$D-CH$_4$ data from different laboratories. Many such intercomparisons have already been made, either on an ad hoc basis or on a more organized scale. However, a systematic evaluation of the underlying calibrations and related measurement offsets among laboratories has been lacking. It is also noted that some measurement programs for $\delta^{13}$C-CH$_4$ and/or $\delta$D-CH$_4$ have been discontinued, and maintaining
access to such datasets including well-established inter-laboratory offsets is important. Here we combine and evaluate the existing comparison results to quantify inter-laboratory measurement differences in order to facilitate the use of $\delta^{13}$C-CH$_4$ and $\delta$D-CH$_4$ data. This study therefore opens the

possibility for merging historic $CH_4$ isotope data reported from multiple laboratories (i.e. synthesis analysis of the existing datasets) for better understanding of the global $CH_4$ budget.

We first present a technical overview of atmospheric $\delta^{13}$C-$CH_4$ and $\delta$D-$CH_4$ measurements and potential causes of measurement offsets among currently available datasets (section 2), and then we summarize

measurement methods by the laboratories that have conducted $\delta^{13}$C-$CH_4$ and $\delta$D-$CH_4$ measurements for air and ice core samples (section 3). In section 4, we report new intercomparison exercises between some groups. We then link the intercomparison results through survey of previous published intercomparisons and provide the current best estimates of measurement offsets among datasets from different laboratories (section 5). Finally, we summarize the current status and briefly discuss possible

causes of the measurement offsets as well as remaining issues that should be kept in mind for combined use of currently existing datasets of isotopic composition of $CH_4$ (section 6).

## 2 Overview of Atmospheric $\delta^{13}$C-$CH_4$ and $\delta$D-$CH_4$ Measurement Techniques

## 2.1 IRMS Measurements for $\delta^{13}$C-$CH_4$ and $\delta$D-$CH_4$

In the 1990s, atmospheric $\delta^{13}$C-$CH_4$ ($\delta$D-$CH_4$) was analyzed using an offline technique in which $CH_4$

was separated from sample air and converted to $CO_2$ ($H_2$) for subsequent offline $\delta^{13}$C-$CH_4$ ($\delta$D-$CH_4$) analyses by dual-inlet isotope ratio mass spectrometry (DI-IRMS) (e.g. Stevens and Rust, 1982; Lowe et al., 1991; Quay et al., 1991, 1999; Sugawara et al., 1996; Poß, 2003). The original methodology was based on the combustion of $CH_4$ in sample air, with interfering compounds such as $CO_2$, $H_2O$, $N_2O$, $CO$ and non-methane hydrocarbons being removed cryogenically, chemically or by gas chromatography

before the $CH_4$ combustion. The number of measurements was limited not only because of laborious and time-consuming laboratory procedures but also because large volumes of air sample were required ($> 100$ $L_{STP}$ for $\delta$D-$CH_4$). Later, a method based on continuous-flow gas chromatography isotope ratio mass spectrometry (GC-IRMS) technique combined with combustion and pyrolysis furnaces became available (Merritt et al. 1995; Burgoyne and Hayes, 1998; Hilkert et al., 1999), which dramatically

reduced time and efforts in the laboratory and likewise the amount of sample air required (now typically 100 $mL_{STP}$). Such systems are now used in most laboratories worldwide for acquiring $\delta^{13}$C-$CH_4$ and

$\delta$D-CH$_4$ data in the current and past atmosphere (Rice et al., 2001; Miller et al., 2002; Sowers et al., 2005; Ferretti et al., 2005; Morimoto et al., 2006; Fisher et al., 2006; Behrens et al., 2008; Umezawa et al., 2009; Brass and Röckmann, 2010; Sperlich et al., 2013; Schmitt et al., 2014; Bock et al., 2014; Brand et al., 2016; Röckmann et al., 2016). Although these systems use a similar measurement principle, they vary in the use of pre-concentration of CH$_4$ in sample air, GC separation, and combustion/pyrolysis, data corrections and in the specific IRMS instrument among laboratories (see Schmitt et al., 2013, section 3 and Table 1). Besides analysis by mass spectrometry, laser-based spectroscopy has also been developed for atmospheric $\delta^{13}$C-CH$_4$ and $\delta$D-CH$_4$ measurements (Bergamaschi et al., 2000; Eyer et al., 2016), but detailed discussion on the technique is beyond the scope of this study.

## 2.2 Standard Scales

VPDB and VSMOW are the standard scales for $\delta^{13}$C-CH$_4$ and $\delta$D-CH$_4$, respectively. To make measurements traceable to these standard scales, each laboratory needs to calibrate its laboratory reference gases against reference materials (RMs) with known values on the standard scales. In this study, the term "calibration" means to measure a laboratory gas (for instance a laboratory working standard gas that is routinely compared with samples) against a standard at higher hierarchy level and to assign to that working standard a $\delta^{13}$C-CH$_4$ or $\delta$D-CH$_4$ value traceable to the standard scale. In principle, all measurements at individual laboratories intend to ultimately anchor their working standards and sample gases to the VPDB or VSMOW scale using the RMs provided by the International Atomic Energy Agency (IAEA) or National Institute of Standards and Technology (NIST) (Coplen et al., 2006; Brand et al., 2014). However, since RMs and recommended calibration methods for measurements of $\delta^{13}$C-CH$_4$ and $\delta$D-CH$_4$ in air have not yet been provided (Sperlich et al., 2012, 2016), individual groups have developed their own calibration strategies.

Since $\delta^{13}$C-CH$_4$ measurement by IRMS is made by $\delta^{13}$C analysis in CO$_2$ oxidized from CH$_4$ in air, some laboratories use pure CO$_2$ gases as a working standard. In many laboratories, these internal CO$_2$ standard gases were calibrated against pure CO$_2$ produced from the primary anchor of the VPDB scale NBS-19 or other RMs by using DI-IRMS (Table 1). Since the typical atmospheric $\delta^{13}$C-CH$_4$ value

(about −47 ‰) differs considerably from the $\delta^{13}C$ value of NBS-19 (+1.95 ‰), some laboratories have used other RMs with VPDB values close to the atmospheric $\delta^{13}C$-CH$_4$ such as LSVEC, IAEA-CO-9 and RM 8563 as a second anchoring point of the VPDB scale (see Table 1) in order to minimize the risk for significant errors in realization of the standard scale (due to scale contraction or $^{17}O$ correction,

described in the following sections). A standard scale established this way at an individual laboratory was often propagated to laboratory-internal CO$_2$ standard gases at lower hierarchy levels, and they were used as the reference in DI-IRMS or GC-IRMS measurement of CO$_2$ processed from CH$_4$ in sample air. Ideally, this accurately links $\delta^{13}C$-CH$_4$ of the sample to the international isotope scale. In contrast, it has been recommended that a measured value of a sample is determined against a reference gas that

undergoes the all preparation steps in the sample measurement line in order to cancel out possible isotopic fractionations due to different treatment between the sample and reference gases (principle of identical treatment; Werner and Brand, 2001). This concept has been taken into account in some laboratories; a working standard is calibrated for $\delta^{13}C$-CH$_4$ and sample measurements are referenced by comparison with measurements of that working standard processed in the same manner (e.g. Brand et

al., 2016). Despite intensions for best traceability to RMs, the variety of calibrations has resulted in diverse realizations of the VPDB scale across $\delta^{13}C$-CH$_4$ measurement programs. As in Table 1, the different RMs that have been applied for $\delta^{13}C$-CH$_4$ calibration include NBS-19 (limestone), IAEA-CO-9 (barium carbonate), LSVEC (lithium carbonate), and RM 8562–8564 (CO$_2$); see Coplen et al. (2006), Brand et al. (2014) and Sperlich et al. (2016). It is also noted that uncertainty of assigned values for

these RMs ranges up to a few tenths ‰ and the assigned values have been revised over time (Brand et al., 2014), which might have complicated the realization of the standard scale at each laboratory. Furthermore, most of these RMs are in different chemical forms, and different isotopic fractionations may have occurred during acid digestion to CO$_2$, which could have biased calibrations at each laboratory. Lastly, WMO (2016) has reported exhaustion of NBS-19 and instability of LSVEC, both of

which are critical RMs for the VPDB scale. Associated possible revision of $\delta^{13}C$ values of RMs in the future will affect consistency of the datasets from different laboratories.

For $\delta D$-CH$_4$, in the conventional offline measurements, CH$_4$ in sample air needs to be processed to H$_2$O followed by reduction to H$_2$ for a subsequent DI-IRMS measurement. GC-IRMS requires pyrolysis of

CH$_4$ to H$_2$. Therefore, individual laboratories have prepared internal standards of H$_2$O (liquid) or H$_2$ (gas), which were calibrated against primary RMs (water) or H$_2$ reference gases certified for $\delta$D (Table 1). Although the situation is less complicated compared to $\delta^{13}$C-CH$_4$ in terms of variety in chemical properties of RMs, the lack of RMs for $\delta$D-CH$_4$ forced laboratories to develop their calibration method

standard scale individually. It is also noted that, similar to $\delta^{13}$C-CH$_4$, principle of identical treatment has not been followed strictly at the all laboratories. If not followed, sample measurements are subject to subtle changes in conditions of the all preparation steps (e.g. conversion of CH$_4$), while such changes do not affect the measured value of a reference gas injected directly to the IRMS.

## 2.3 Scale Contraction

It has been found that cross contamination between sample and reference CO$_2$ gases shrinks the $\delta^{13}$C distance measured on DI-IRMS (Meijer et al., 2000; Verkouteren et al., 2003a, 2003b). This effect is known as scale contraction or $\eta$ effect, and the magnitude is specific to the IRMS instrument and its settings. Since the VPDB scale for $\delta^{13}$C-CH$_4$ has been realized and propagated via CO$_2$ calibrations by DI-IRMS at individual laboratories, the instrument-dependent scale contraction effect could have

caused a significant difference in measurement values, especially at the low $\delta^{13}$C values of atmospheric CH$_4$ of about $-47$ ‰ (Wendeberg et al., 2013).

## 2.4 [17]O Correction

For measurement of $\delta^{13}$C-CH$_4$ by IRMS, CH$_4$ is first oxidized to CO$_2$ and the different isotopic variants of the produced CO$_2$ are then registered on Faraday cups with mass to charge ratios $m/z$ of 44, 45 and

46. Since the raw ion beam intensity for $m/z = 45$ is the sum of $^{13}$C$^{16}$O$_2$ and $^{12}$C$^{17}$O$^{16}$O, the final $\delta^{13}$C value is obtained by correcting for the contribution of the $^{17}$O containing isotopologue, known as $^{17}$O correction (e.g. Assonov and Brenninkmeijer, 2003). Several algorithms such as Craig (1957) and Santrock et al. (1985) have been suggested (see Assonov and Brenninkmeijer (2003) and references therein) and implemented into software/programs of the IRMS companies and individual laboratories.

Assonov and Brenninkmeier (2003) showed that the bias caused by different $^{17}$O-correction algorithms

could exceed general repeatability achieved by IRMS measurements. The $^{17}O$ correction method of each laboratory is listed in Table 1.

## 2.5 Krypton Interference in GC-IRMS

The transition from DI-IRMS to GC-IRMS analyses reduced the analytical effort, but also introduced complications that were initially not recognized and taken into account. It was recently found that atmospheric krypton (Kr) interferes with the $\delta^{13}C$-$CH_4$ GC-IRMS analysis if Kr is present in the ion source during the data acquisition of the $CO_2$ peak generated from $CH_4$ oxidation (hereafter $CH_4$-derived $CO_2$ peak) (Schmitt et al., 2013). Thus the $\delta^{13}C$-$CH_4$ measurements on a GC-IRMS system can be biased if Kr is not separated sufficiently from either $CH_4$ or from the $CH_4$-derived $CO_2$ peak after the $CH_4$ combustion. Schmitt et al. (2013) demonstrated that the doubly charged krypton isotope $^{86}Kr^{2+}$, produced in the ion source of an IRMS, can cause lateral tailing extending into the Faraday cups used for $\delta^{13}C$ analysis (i.e. *m/z* of 44, 45 and 46), which compromises the measured signal of the $CH_4$-derived $CO_2$ peak. This effect had not been recognized for more than a decade since the early years of GC-IRMS measurements (Merritt et al., 1995), and thus has not been taken into account in many datasets of atmospheric $\delta^{13}C$-$CH_4$ reported in the meantime (e.g. Miller et al., 2002; Morimoto et al., 2006; Fisher et al., 2011; Röckmann et al., 2011; Umezawa et al., 2012a, 2012b). Furthermore, because the Kr effect is system-dependent and variable with time (Schmitt et al., 2013), applying plausible corrections to past data may not be feasible. Likewise, several gas species including Kr can affect $\delta D$-$CH_4$ measurements, and this effect is also system-dependent (Bock et al., 2014).

Several solutions have been suggested to eliminate or account for the Kr interference (Schmitt et al., 2013). Among them, three methods have been implemented at different laboratories (Table 1). Briefly, (1) After the $CH_4$ oxidation to $CO_2$, Kr is separated from the $CH_4$-derived $CO_2$ by using a post combustion separation column (PCS) or cryogenically. (2) An offset due to the Kr interference is estimated by comparison with a DI-IRMS measurement (DI-offset). (3) The Kr interference peak is subtracted from the raw ion current time series of the IRMS acquisition (raw ion current correction). More detailed description has been presented in Schmitt et al. (2013).

## 3. Measurements of Participating Laboratories

In this section, we briefly document measurement systems of individual laboratories for ease of reference in the following intercomparisons (sections 4 and 5). For details, we refer to more dedicated publications listed in Table 1. The table also visualizes differences among laboratories in terms of possible causes of the measurement offsets described in section 2.

### 3.1 NIWA

National Institute for Water and Atmospheric Research (NIWA, originally INS (Institute of Nuclear Sciences), later INGS (Institute of Nuclear and Geological Sciences) until 1992) successfully initiated systematic measurements of atmospheric $\delta^{13}$C-CH$_4$ by means of offline CH$_4$ separation and conversion followed by a DI-IRMS measurement in 1988 (Lowe et al., 1988, 1991). A suite of CO$_2$ working gases with $\delta^{13}$C-CH$_4$ values around −47 ‰ referenced to IAEA materials were utilized to calibrate the measurements. An overall reproducibility of the $\delta^{13}$C-CH$_4$ measurement was evaluated to be 0.02 ‰ (Lowe et al., 1991). The $\delta^{13}$C-CH$_4$ measurements at NIWA are ultimately calibrated against CO$_2$ produced from NBS-19, IAEA-CO-9 and LSVEC. The long-term $\delta^{13}$C-CH$_4$ records have been presented since then (Lowe et al., 1994, 1997, 2004; Bergamaschi et al., 2001; Schaefer et al., 2016). Bromley et al. (2012) reported that repeated measurements of the two working reference gases and archived air indicated no detectable drift over 16 years since 1992. NIWA also operates a GC-IRMS system since 2004 (Ferretti et al., 2005) with reproducibility of 0.1 ‰. The Kr interference on the GC-IRMS $\delta^{13}$C-CH$_4$ measurement has been identified, which is corrected by an offset relative to the conventional DI-IRMS measurement (see section 4.1).

### 3.2 IMAU

The GC-IRMS system at the Institute for Marine and Atmospheric research Utrecht (IMAU) has been described by Brass and Röckmann (2010). The measurement reproducibility is estimated to be 0.07 ‰ and 2.3 ‰ for $\delta^{13}$C-CH$_4$ and $\delta$D-CH$_4$, respectively. Sample air is measured against reference air that is processed in the GC-IRMS system in the same manner as a sample. The IMAU $\delta^{13}$C-CH$_4$ standard scale is based on a set of assigned values for 13 firn air samples measured at Max Planck Institute for

Chemistry (MPIC) (Bräunlich et al., 2001) and they are ultimately referenced to a $CO_2$ gas produced from NBS-19 (Röckmann, 1998; Bergamaschi et al., 2000). The $\delta D\text{-}CH_4$ standard scale is based on a set of reference gases originally produced at MPIC (see section 2.3). These calibration details have been documented also by Sperlich et al. (2016). The IMAU system was originally affected by Kr but later modified to remove this interference. A correction was applied for data obtained before the system modification (Schmitt et al., 2013).

### 3.3 MPIC

MPIC has reported $\delta^{13}C\text{-}CH_4$ and $\delta D\text{-}CH_4$ measurements at a baseline station (Bergamaschi et al., 2000) and for firn air samples (Bräunlich et al., 2001) based on an offline DI-IRMS measurement for $\delta^{13}C\text{-}CH_4$ (Bergamaschi et al., 2000) and a tunable diode laser based absorption spectrometer (TDLAS) for $\delta D\text{-}CH_4$ (Bergamaschi et al., 1994). Some firn air measurements by Bräunlich et al. (2001) were performed by using a GC-IRMS system at Laboratory of Glaciology and Geophysics of the Environment. As described in section 3.2, the $\delta^{13}C\text{-}CH_4$ and $\delta D\text{-}CH_4$ standard scales of MPIC are basis of that of IMAU. For the $\delta^{13}C\text{-}CH_4$ DI-IRMS measurement, the $CH_4$-derived $CO_2$ was measured against a working standard (pure $CO_2$) that was calibrated against NBS-19 on a DI-IRMS system (Röckmann, 1998; Bergamaschi et al., 2000). The MPIC $\delta D\text{-}CH_4$ scale is based on measurements of standard gases at the Bundesanstalt für Geowissenschaften und Rohstoffe, Hannover, Germany; $CH_4$ was combusted to $CO_2$ and $H_2O$, followed by reduction of $H_2O$ to $H_2$ for subsequent DI-IRMS analysis on $H_2$; the calibration was made against VSMOW and SLAP (Bergamaschi et al., 2000). The measurements of atmospheric $\delta^{13}C\text{-}CH_4$ and $\delta D\text{-}CH_4$ at MPIC were discontinued.

### 3.4 MPI-BGC

Max Planck Institute for Biogeochemistry (MPI-BGC) set up a GC-IRMS system for $\delta^{13}C\text{-}CH_4$ and $\delta D\text{-}CH_4$ measurements, and it has been operated for air samples collected at baseline stations (Brand et al., 2016). The long-term (3 years) reproducibility was assessed to be 0.12 ‰ for $\delta^{13}C\text{-}CH_4$ and 1.0 ‰ for $\delta D\text{-}CH_4$. Initially, the GC-IRMS measurements had been anchored to a working standard air calibrated by IMAU. The Kr effect was eliminated by a PCS column, and the initial calibration has in the

meantime been replaced by a new primary calibration, where measurements are ultimately anchored to NBS-19 and LSVEC for $\delta^{13}$C-CH$_4$ and VSMOW-2 and SLAP-2 for $\delta$D-CH$_4$ (Sperlich et al., 2016). This calibration, termed JRAS-M16, is the basis for the $\delta^{13}$C-CH$_4$ and $\delta$D-CH$_4$ values from MPI-BGC reported in this manuscript.

## 3.5 UCI

University of California Irvine (UCI) measured atmospheric $\delta^{13}$C-CH$_4$ by offline DI-IRMS and $\delta$D-CH$_4$ by GC-IRMS (Tyler et al., 1999, 2007; Kai et al., 2011). The UCI GC-IRMS system for both $\delta^{13}$C-CH$_4$ and $\delta$D-CH$_4$ has been described in detail by Rice et al. (2001). The measurement reproducibility of the GC-IRMS system was estimated to be 0.05 ‰ and 1.5 ‰ for $\delta^{13}$C-CH$_4$ and $\delta$D-CH$_4$, respectively, while that of the offline DI-IRMS $\delta^{13}$C-CH$_4$ measurement was 0.05 ‰. Samples were measured against laboratory working standard gases of pure CO$_2$ for $\delta^{13}$C-CH$_4$ and pure H$_2$ for $\delta$D-CH$_4$. The $\delta^{13}$C-CH$_4$ calibration is based on a CO$_2$ reference gas provided by NIWA, which was compared with CO$_2$ produced from NBS-19 and IAEA-CO-9 (Lowe et al., 1999). The $\delta$D-CH$_4$ calibration is referenced to three H$_2$ gas cylinders purchased from Oztech Gas Company (Rice et al., 2001). The possible Kr interference on the GC-IRMS system is unclear (the laboratory is now closed), but it appears that the Kr effect had been avoided using liquid nitrogen cooling of the GC column as surmised by Schmitt et al. (2013).

## 3.6 TU

The GC-IRMS system at Tohoku University (TU) has been described by Umezawa et al. (2009). The measurement reproducibility is estimated to be 0.08 ‰ for $\delta^{13}$C-CH$_4$ and 2.2 ‰ for $\delta$D-CH$_4$. Sample measurements are made against pure CO$_2$ and H$_2$ working standard gases for $\delta^{13}$C-CH$_4$ and $\delta$D-CH$_4$, respectively. The $\delta^{13}$C-CH$_4$ calibration is based on a CO$_2$ primary gas produced from NBS-19. The H$_2$ working standard for $\delta$D-CH$_4$ measurement is referenced to water laboratory standards that are calibrated against VSMOW and SLAP. Measured $\delta$D-CH$_4$ values are corrected so that the value of a laboratory test gas is kept constant over time to take into account fluctuations in the measured value due to the condition of the pyrolysis furnace (Umezawa et al., 2009, 2012a). The Kr interference on the

$\delta^{13}$C-CH$_4$ measurement was identified, but modification or correction has not been implemented. It has been documented that $\delta^{13}$C-CH$_4$ measurement at TU shifted by +0.27 ‰ after July 2008 (the cause of this sudden shift has yet to be identified) and measurements afterwards were corrected for this value to keep the data consistency (Umezawa et al., 2012a, 2012b). Note that TU made rigorous re-evaluation of the long-term measurements of their working standard gas recently, and the TU $\delta^{13}$C-CH$_4$ datasets will be revised accordingly. Therefore, the comparison numbers presented here are not comparable to those for earlier publications (Umezawa et al., 2009, 2011, 2012a, 2012b).

### 3.7 NIPR

National Institute of Polar Research (NIPR) reported $\delta^{13}$C-CH$_4$ measurements at an Arctic site using a GC-IRMS system (Morimoto et al., 2006). The measurement reproducibility was evaluated to be 0.06 ‰. The $\delta^{13}$C-CH$_4$ calibration follows same procedure as TU. By injecting different quantities of Kr, it was confirmed that ambient Kr does not significantly interfere with the $\delta^{13}$C-CH$_4$ measurements at NIPR.

### 3.8 UW

University of Washington (UW) reported extensive $\delta^{13}$C-CH$_4$ and $\delta$D-CH$_4$ measurements using an offline DI-IRMS system (Quay et al., 1991, 1999). The reproducibility was estimated to be 0.1 ‰ for $\delta^{13}$C-CH$_4$ and 3–4 ‰ for $\delta$D-CH$_4$. The $\delta^{13}$C-CH$_4$ calibration is based on measurements against NBS-19 (Quay et al., 1999), while the earlier measurements were calibrated against NBS-20 and NBS-16 (Quay et al., 1991). The $\delta$D-CH$_4$ was anchored to calibration by VSMOW and SLAP. Systematic measurements of air standards showed that no significant time shift (+0.001±0.002 ‰ yr$^{-1}$) affected their $\delta^{13}$C-CH$_4$ dataset for 1988–1995 (Quay et al., 1999).

### 3.9 UHEI

University of Heidelberg (UHEI) carried out $\delta^{13}$C-CH$_4$ measurements by DI-IRMS (Levin et al., 1999, 2012). The typical measurement reproducibility was evaluated to be 0.05 ‰ (Levin et al., 1999). The UHEI $\delta^{13}$C-CH$_4$ measurements are calibrated against CO$_2$ reference materials (RM 8562, RM 8563 and RM 8564) (Behrens et al., 2008). Although reported previously only for signatures of source CH$_4$

(Levin et al., 1993), UHEI also made offline $\delta$D-CH$_4$ measurements on atmospheric samples by DI-IRMS and TDLAS (Poß, 2003). The $\delta$D-CH$_4$ measurements by DI-IRMS were made on pure H$_2$ (H$_2$O from CH$_4$ oxidation converted to H$_2$ with zinc as catalyst) and were calibrated against VSMOW and SLAP. Note that UHEI recently re-evaluated all their atmospheric $\delta^{13}$C-CH$_4$ and $\delta$D-CH$_4$ measurements

rigorously, based on the history of laboratory standards used; therefore, comparison numbers published in earlier works are not comparable to the revised values presented here.

## 3.10 INSTAAR

Institute of Arctic and Alpine Research (INSTAAR) of the University of Colorado Boulder has measured $\delta^{13}$C-CH$_4$ and, intermittently, $\delta$D-CH$_4$ using a GC-IRMS system for flask air samples from

the cooperative sampling network of National Oceanic and Atmospheric Administration (NOAA) (Miller et al., 2002). Reproducibilities of the $\delta^{13}$C-CH$_4$ and $\delta$D-CH$_4$ measurements are evaluated to be 0.08 ‰ and 2 ‰, respectively (Miller et al., 2002; White et al., 2016). The INSTAAR $\delta^{13}$C-CH$_4$ measurement currently follows the UCI calibration, while the $\delta$D-CH$_4$ measurement is not explicitly anchored to the VSMOW scale (White et al., 2016). The Kr interference on the $\delta^{13}$C-CH$_4$ measurement

is significant, and a PCS column was therefore implemented into the system in May 2017. Correction of the data for the Kr interference (1998–present) is under evaluation. Of the data presented here, only the ice core intercomparison round robin (section 3.4) and the INSTAAR-MPI-BGC comparison (section 3.5) have not been interfered by Kr.

## 3.11 RHUL

Royal Holloway University of London (RHUL) measured atmospheric $\delta^{13}$C-CH$_4$ using an offline DI-IRMS technique (Lowry et al., 2001) and a GC-IRMS system (Fisher et al., 2006, 2011; Nisbet et al., 2016). Reproducibility of the DI-IRMS measurement was evaluated to be 0.04 ‰ (Lowry et al., 2001) and that by the GC-IRMS is 0.05 ‰ (Fisher et al., 2006). They made $\delta^{13}$C-CH$_4$ calibrations ultimately to IAEA carbonate materials NBS-19 and IAEA-CO-9 (Lowry et al., 2001; Fisher et al., 2006). Note

that RHUL applies an offset correction of −0.20 ‰ for the measured value by GC-IRMS (sections 4.6 and 5.11).

### 3.12 PDX

Portland State University (PDX) reported $\delta^{13}$C-CH$_4$ and $\delta$D-CH$_4$ measurements for archive air samples (Rice et al., 2016). The PDX measurement system has been described in Teama (2013) with some updates since Rice et al. (2001). The $\delta^{13}$C-CH$_4$ and $\delta$D-CH$_4$ reproducibilities are 0.07 ‰ and 2.0 ‰, respectively, and PDX shares the standard scales with UCI for both $\delta^{13}$C-CH$_4$ and $\delta$D-CH$_4$ (Rice et al., 2016).

### 3.13 PSU

Pennsylvania State University (PSU) reported $\delta^{13}$C-CH$_4$ and $\delta$D-CH$_4$ data from ice cores and firn air using a GC-IRMS system (e.g. Sowers et al., 2005; Sowers, 2010). The overall measurement reproducibility including every step for ice core measurements was evaluated to be 0.3 ‰ for $\delta^{13}$C-CH$_4$ and 3 ‰ for $\delta$D-CH$_4$ (Sowers, 2010). The PSU $\delta^{13}$C-CH$_4$ measurements are calibrated against $CO_2$ RMs (RM 8563 and RM 8564). The $\delta$D-CH$_4$ calibration is made against $H_2$ gas bottles from Oztech Gas Company (Sowers, 2006).

### 3.14 UB

University of Bern (UB) makes $\delta^{13}$C-CH$_4$ measurements from ice cores using a GC-IRMS system with an overall reproducibility of 0.15 ‰ (Schmitt et al., 2014; Bock et al., 2017). The UB measurements are referenced to a whole-air working standard with CH$_4$ mole fraction of 1508.2 ppb and an assigned $\delta^{13}$C-CH$_4$ value of −47.34±0.02 ‰ (named "Boulder, CA08289" in Schmitt et al., 2014). This value is anchored to the standard scale used at INSTAAR (section 3.10). UB also measures $\delta$D-CH$_4$ for ice core samples (Bock et al., 2010, 2014, 2017). The overall measurement precision for ice core sample (including extraction of air from an ice sample) was evaluated to be 2.3 ‰. The UB $\delta$D-CH$_4$ measurement is referenced by using an ambient air cylinder (named "Air Controlé") with a $\delta$D-CH$_4$ value of −93.6±2.8 ‰, which was cross-referenced to a high pressure cylinder filled at the Alert Station ("Alert 2002/11" with $\delta$D-CH$_4$ of −82.2±1.0 ‰) analyzed on the scale maintained at UHEI (Bock et al., 2010, 2014). However, this value has to be corrected to −85.2±1.0 ‰ to account for the recent re-evaluation at UHEI (section 3.9). All UB data published after 2011 are free of Kr interference.

### 3.15 AWI

Alfred Wegener Institute Helmholtz Centre for Polar and Marine Research (AWI) reported $\delta^{13}$C-CH$_4$ measurements from ice cores using a GC-IRMS system (Behrens et al., 2008; Fischer et al., 2008; Möller et al., 2013). The measurement reproducibility was estimated to be 0.2 ‰. The $\delta^{13}$C-CH$_4$
measurements employed the UHEI standard scale via comparison of measurements of an air sample collected at Neumayer Station, Antarctica (Möller et al., 2013).

### 3.16 CIC

Centre for Ice and Climate (CIC) of the Niels Bohr Institute has reported $\delta^{13}$C-CH$_4$ measurements from ice cores (Sperlich et al., 2015) using a GC-IRMS system with measurement reproducibility of 0.09 ‰
(Sperlich et al., 2013). CIC also set up an offline combustion system for samples with large amount of CH$_4$, which is combined with DI-IRMS for $\delta^{13}$C-CH$_4$ and with either a high Temperature Conversion/Elemental Analyser (TC/EA) coupled to IRMS or laser spectroscopy for $\delta$D-CH$_4$ (Sperlich et al., 2012); the measurement reproducibility is 0.04 ‰ for $\delta^{13}$C-CH$_4$ and 0.7 ‰ for $\delta$D-CH$_4$. The CIC measurements are referenced to RM 8563 for $\delta^{13}$C-CH$_4$ and VSMOW-2 and SLAP-2 for $\delta$D-CH$_4$. The
combined uncertainty of this analytical system including the uncertainty of the entire traceability chain was estimated as 0.07 ‰ for $\delta^{13}$C-CH$_4$ and 0.7 ‰ for $\delta$D-CH$_4$ (Sperlich et al., 2016).

### 4 Intercomparison Exercises

### 4.1 Intercomparison between UCI and IMAU

An intercomparison between UCI and IMAU was made by analyzing 6 air samples at both laboratories;
the air samples were collected along a flight track of commercial aircraft in the upper troposphere in the early phase of the CARIBIC (Civil Aircraft for the Regular Investigation of the atmosphere Based on an Instrument Container) project (Brenninkmeijer et al., 1999). The original samples were collected into large stainless steel cylinders (21 L) and aliquots of them were transferred into smaller stainless steel canisters (~2.3 L) for storage after delivery to the MPIC laboratory. Different sub samples from
identical original samples were sent to UCI and IMAU for analyses, and they were measured at UCI in

2008 and at IMAU in 2012 to 2013. The measurement results at both laboratories are summarized in Table 2. The result indicated significant differences of +0.42±0.04‰ for $\delta^{13}$C-CH$_4$ (UCI value is higher than IMAU) and of −10.7±0.7‰ for $\delta$D-CH$_4$ (UCI value is lower than IMAU).

## 4.2 Intercomparison between TU/NIPR and IMAU

An intercomparison between TU/NIPR and IMAU was carried out during 2013–2015. The TU laboratory prepared four stainless steel canisters (~1 L) filled with dried ambient air (canisters MD1 and MD2) and CH$_4$-in-synthetic air gas (canisters MD3 and MD4) with CH$_4$ mole fractions ranging from 899 to 2117 ppb on the TU CH$_4$ scale (Aoki et al., 1992; Umezawa et al., 2014) (Table 3). The canisters were analyzed at TU and then sent to IMAU, after which they were sent back to TU and reanalyzed to

confirm the stability of the air samples in the canisters during the intercomparison exercise. The measurements at TU before and after the transport to IMAU throughout April 2013 to July 2015 indicated that possible drifts during canister storage and transportation are small (< 0.1 ‰ for $\delta^{13}$C-CH$_4$ and < 3.5 ‰ for $\delta$D-CH$_4$). NIPR also measured the canisters for $\delta^{13}$C-CH$_4$. The results indicate significant differences of +0.50±0.07 ‰ for $\delta^{13}$C-CH$_4$ (TU value is higher than IMAU) and of

−13.9±0.9 ‰ for $\delta$D-CH$_4$ (TU value is lower than IMAU) (Table 3). The measurements of the four canisters at NIPR were +0.48±0.11 ‰ higher than IMAU. However, the differences of $\delta^{13}$C-CH$_4$ measurements are smaller for the ambient air samples (MD1 and MD2) than the CH$_4$-in-synthetic air samples (MD3 and MD4). It is also noted that the $\delta^{13}$C-CH$_4$ difference between the laboratories is largest for the low CH$_4$ mole fraction (~900 ppb) sample (MD3). The cause is unclear, but might be

related to (1) deviation in $\delta^{13}$C-CH$_4$ of the latter samples from the typical atmospheric value, i.e., scale contraction effect, (2) difference in air matrix, i.e., natural versus synthetic air and (3) difference in linearity with respect to CH$_4$ mole fraction. This result therefore indicates that the measurement offset is not constant for a wide range of $\delta^{13}$C-CH$_4$ values and CH$_4$ mole fractions as well as differences in air matrix. Since we focus in this study on comparison for atmospheric samples, the intercomparison

results for the ambient air samples are considered as inter-laboratory measurement offsets. The average differences for ambient air are +0.40±0.04 ‰ for TU and +0.31±0.03 ‰ for NIPR relative to IMAU. Likewise, the $\delta$D-CH$_4$ offset of TU versus IMAU is considered to be −13.1±0.6 ‰.

## 4.3 Intercomparison between UHEI and MPI-BGC

An intercomparison between UHEI and MPI-BGC was conducted in 2013 on six archived air samples from Neumayer station, Antarctica. These samples, collected in the time period from 1988 to 2008 had been analyzed by UHEI for $\delta^{13}$C-CH$_4$ and $\delta$D-CH$_4$ by DI-IRMS (two samples were analyzed for $\delta$D-CH$_4$ additionally by TDLAS) during 2003–2010 and were stored in high-pressure cylinders. The typical reproducibility for the measurements is between 0.02 and 0.05 ‰ for $\delta^{13}$C-CH$_4$ and between 1.6 and 2.6 ‰ for $\delta$D-CH$_4$. In 2013, duplicate aliquots were sampled in 1-L glass flasks and analyzed at MPI-BGC. The measurement results at both laboratories are summarized in Table 4. The results show insignificant measurement offsets of +0.02±0.05 ‰ for $\delta^{13}$C-CH$_4$ and of +0.4±0.6 ‰ for $\delta$D-CH$_4$ (with the MPI-BGC values being more negative than those from UHEI in both cases).

## 4.4 Round Robin Comparison among Ice Core Analysis Laboratories

A round robin cylinder exercise was initiated to facilitate intercomparison of those laboratories who measure $\delta^{13}$C-CH$_4$ and $\delta$D-CH$_4$ in ice core and firn air samples. Part of this exercise has been presented previously (Table 2 in Schmitt et al., 2013). Three high-pressure Al cylinders were filled with varying trace gas composition to mimic present day, pre-industrial and last-glacial air mole fractions. The CH$_4$ mole fractions of these cylinders were 1830.6 ppb (CA 03560), 904.0 ppb (CC 71560) and 372.2 ppb (CA 01179) on the NOAA-2004 CH$_4$ scale (Dlugokencky et al., 2005), respectively. The cylinders were shipped to the laboratories listed in Table 5 for analyses of all constituents that each lab was capable of measuring at that time. In Table 5, we list the $\delta^{13}$C-CH$_4$ and $\delta$D-CH$_4$ results from each laboratory. The Kr interfering artefact associated with GC-IRMS $\delta^{13}$C-CH$_4$ analyses was taken into account in many of the analyses (Schmitt et al., 2013). In some cases, aliquots from the tanks were measured using offline combustion to CO$_2$ followed by $\delta^{13}$C-CH$_4$ analyses via conventional DI-IRMS. The cylinders were remeasured at PSU at the end of the round robin to verify that the isotopic composition had not shifted over the 9 years during the transportation of the cylinders. The difference between the 2007 and 2016 $\delta^{13}$C-CH$_4$ measured at PSU were less than 0.14 ‰ for two of the three cylinders, indicating that the isotopic composition of the cylinder air was stable throughout the intercomparison exercise. The third cylinder (CA 01179) was 0.58 ‰ off from the original measurement, which is just outside the analytical

uncertainty associated with PSU measurements. There may have been slight drift over the 9 years between measurements, although the cause has yet to be resolved. The results of the $\delta^{13}$C-CH$_4$ intercomparison showed agreement with the average standard deviation amongst all six participating laboratories better than 0.37 ‰ for the cylinders with high (CA 03560) and middle (CC 71560) mole fractions. $\delta$D-CH$_4$ results show more scatter due to the difficult nature of the measurements and the offset among the standard scales.

## 4.5 Intercomparison between INSTAAR and MPI-BGC

An intercomparison between INSTAAR and MPI-BGC was recently made by analyzing three air cylinders at both laboratories. They were measured at MPI-BGC between April and July of 2016 and at INSTAAR between May and June of 2017. Two of the cylinders have ambient CH$_4$ mole fraction (~1900 ppb; HUEY-001 and DEWY-001) and the other has a lower value (~1500 ppb; LOUI-001) (Table 6). In addition, air from another suite of cylinders was sampled into flasks at INSTAAR and sent to MPI-BGC. Measurements at MPI-BGC and INSTAAR were made in January–February of 2017 and May–June of 2017, respectively. The four cylinders (CART-001, STAN-001, KENN-001 and KYLE-001) have different CH$_4$ mole fractions and $\delta^{13}$C-CH$_4$ values. The measurement results are summarized in Table 6. The INSTAAR data presented here were not interfered by Kr by installing a PCS column into the system. The results show significant but consistent measurement offsets of +0.28±0.01 ‰ for the five cylinders with different CH$_4$ mole fractions and ambient $\delta^{13}$C-CH$_4$ values (with the INSTAAR values being more positive than those from MPI-BGC). The measurements for the cylinder with low $\delta^{13}$C-CH$_4$ value were 0.60 ‰ off between both laboratories presumably due to the scale contraction effect. It is noted that the INSTAAR measurements without the Kr removal had yielded a higher $\delta^{13}$C-CH$_4$ value (+0.44±0.02 ‰ relative to the MPI-BGC measurement) for one cylinder (LOUI-001), which presumably reflects the Kr interference pronounced at lower CH$_4$ mole fraction.

## 4.6 Intercomparison based on co-located samples through the NOAA cooperative sampling network

The Cooperative Flask Sampling Network, operated by the NOAA Global Monitoring Division, collects air samples from numerous sites around the world, and INSTAAR has analyzed those air samples for

$\delta^{13}$C-CH$_4$ since 1998. There are several sites where air sample collections by other laboratories have been made concurrently. RHUL has analyzed air samples at Alert (ALT), Canada and Ascension Island (ASC), and NIWA has done at Baring Head (BHD), New Zealand. Although the individual laboratories do not measure the same sample air in these cases, these co-located air samples provide an opportunity for assessment of possible measurement offsets as examined previously (Levin et al., 2012). (1) For the RHUL-INSTAAR difference, the $\delta^{13}$C-CH$_4$ data at ALT during 2009–2014 and at ASC during 2010–2015 were compared to each other if both air samples were collected within a 10 hour interval. The ALT and ASC comparisons indicated that the INSTAAR measurement is +0.05±0.16 ‰ ($N$=350) and 0.00±0.17 ‰ ($N$=80) higher than RHUL, respectively. Note that, for this comparison, the RHUL GC-IRMS data were corrected by −0.20 ‰; the offset value was estimated from measurements of flasks filled from two different cylinders (CH$_4$ in air, both at ambient mole fraction level, one at ambient $\delta^{13}$C-CH$_4$ and the other at about −56 ‰ by spiking $^{13}$C-depleted CH$_4$). (2) For the NIWA-INSTAAR comparison, the $\delta^{13}$C-CH$_4$ data at BHD during 2009–2014 from both laboratories were compared if both air samples were collected within a 15 hour interval. The result indicates that the INSTAAR measurement is +0.08±0.11 ‰ ($N$=45) higher than NIWA.

## 5 Measurement offsets among laboratories

Here we revisit intercomparisons published previously. Some laboratories employed a standard scale from another laboratory. Such intercomparisons and inter-laboratory scale propagations reported in the literature are displayed in Fig. 1. In this section we review the previous and present intercomparison measurements and accordingly suggest plausible measurement offsets among different laboratories (Fig. 2). Relevant information is summarized in Table 1 and the subsections below correspond to those in section 3. Since some laboratories focus on $\delta^{13}$C-CH$_4$ and $\delta$D-CH$_4$ measurements from ice core and firn air samples to elucidate changes of atmospheric CH$_4$ in the past, Fig. 2 also combines $\delta^{13}$C-CH$_4$ and $\delta$D-CH$_4$ data both for the modern and past atmosphere. It is however noted that Fig. 2 suggests the measurement offsets at the modern CH$_4$ mole fraction and isotopic ratios and that such values could be different for the past atmosphere (see sections 4.2, 4.4 and 4.5).

In this study, we report $\delta^{13}$C-CH$_4$ offsets with respect to the conventional DI-IRMS measurement at NIWA (Lowe et al., 1991) because NIWA's $\delta^{13}$C-CH$_4$ measurements have been compared with those from most laboratories to date (Table 1 and Fig. 1). In contrast, $\delta$D-CH$_4$ measurements from different laboratories have been limited. We report $\delta$D-CH$_4$ offsets of different laboratories with respect to the

IMAU measurement. The uncertainties presented in this study are generally standard errors of the mean, but numbers in the literature are cited as is. It should be therefore noted that the uncertainties, in particular those calculated by error propagation, are not rigorously consistent at all places in the manuscript.

### 5.1 NIWA

$\delta^{13}$C-CH$_4$: As listed in Table 1, the DI-IRMS measurement at NIWA has been repeatedly intercompared with other laboratories. Importantly for this comparison, Bromley et al. (2012) reported the long-term stability of the measurement over the years 1992–2007, and it is likewise confirmed until 2011. The NIWA GC-IRMS system, based on the methodology of Miller et al. (2002), has an offset relative to the DI-IRMS of −0.19±0.26 ‰. Measurements on the GC-IRMS informing this instrument comparison are

subject to the Kr interference. A Kr-correction has since been derived in an empirical equation from the round robin intercomparison results (Schmitt et al., 2013 and section 4.4), accounting for differences in CH$_4$ mole fraction and an exponential fit to the GC-IRMS versus DI-IRMS results. The GC-IRMS system is currently equipped with a PCS column to eliminate the Kr interference.

### 5.2 IMAU

$\delta^{13}$C-CH$_4$: According to Schmitt et al. (2013), the IMAU measurement at the present CH$_4$ mole fraction level is in agreement to NIWA with an offset value of −0.04±0.07 ‰ (No. 2 in Fig. 2a). This corresponds to the round robin comparison for the cylinder with CH$_4$ mole fraction of 1830.6 ppb (CA 03560) in Table 5 (section 4.4). The difference is −0.03±0.05 ‰ for data analyzed before the modification to remove the Kr interference (see Table 2 in Schmitt et al. (2013)). The intercomparison

in this study (section 3.4) also shows that the IMAU offset is −0.08±0.11 ‰ for the cylinder with the CH$_4$ mole fraction of 904.0 ppb (CA 71560).

$\delta$D-CH$_4$: As listed in Table 1, IMAU has made most intercomparisons with other laboratories so far. It is noted that the standard scale at IMAU was propagated from MPIC (Bergamaschi et al., 2000; section 2.2), and that it recently showed a reasonable agreement with the recent calibration at MPI-BGC (Sperlich et al., 2016).

## 5.3 MPIC

$\delta^{13}$C-CH$_4$: As written in section 3.3, the standard scale at MPIC was transferred to IMAU (Brass and Röckmann, 2010; Sperlich et al., 2016). Since no direct comparison with NIWA is available, the MPIC offset relative to NIWA is estimated to be −0.04±0.07 ‰, identical to the IMAU offset (No. 3 in Fig. 2a).

$\delta$D-CH$_4$: Bock et al. (2010) reported an intercomparison using firn air samples between UB and MPIC, which indicated that, combined with the UB $\delta$D-CH$_4$ offset (section 5.14), the MPIC $\delta$D-CH$_4$ offset is +0.3±1.1 ‰ with respect to IMAU (No. 3 in Fig. 2b).

## 5.4 MPI-BGC

$\delta^{13}$C-CH$_4$: Sperlich et al. (2016) quantified the offset of the IMAU standard scale relative to the primary standard scale at MPI-BGC. It was indicated that the MPI-BGC measurement differs by −0.03±0.10 ‰ from the IMAU standard scale. Combined with the IMAU offset relative to NIWA (section 5.2), the MPI-BGC offset is estimated to be −0.07±0.12 ‰ (No. 4 in Fig. 2a).

$\delta$D-CH$_4$: According to Sperlich et al. (2016), the MPI-BGC measurement is −4.2±1.2 ‰ from IMAU (No. 4 in Fig. 2b).

## 5.5 UCI

$\delta^{13}$C-CH$_4$: Intercomparison exercises of UCI with external laboratories have been made several times. The oldest intercomparison (Lowe et al., 1991) reported good agreement (< 0.02 ‰) between the former UCI laboratory (S. Tyler at NCAR) and NIWA (INS, IGNS at that time). Among the later measurements, there are two direct intercomparisons with NIWA. (1) Tyler et al. (2007) reported an intercomparison result of UCI to be −0.01±0.09 ‰ with respect to NIWA (No. 5 left in Fig. 2a). For

this comparison, 16 air samples collected at Niwot Ridge, Colorado or Baring Head, New Zealand were exchanged between UCI and NIWA in 1998–1999. (2) This study (section 4.4 and Table 5) shows that the UCI measurement is +0.14±0.12 ‰ (No. 5 middle in Fig. 2a) and +0.04±0.08 ‰ higher than NIWA for the cylinders with high (CA 03560) and middle (CC 71560) $CH_4$ mole fractions, respectively. (3) In contrast, the intercomparison in this study (section 4.1 and Table 2) combined with the IMAU offset (section 5.2) yields +0.42±0.04 ‰ relative to NIWA (not shown in Fig. 2a), inconsistent with the above intercomparison results made earlier. The determinate error has yet to be resolved.

$\delta$D-$CH_4$: According to the intercomparison in this study (section 4.1), the UCI has a $\delta$D-$CH_4$ offset of −10.7±0.7 ‰ with respect to IMAU (No. 5 in Fig. 2b).

## 5.6 TU

$\delta^{13}$C-$CH_4$: The intercomparison in this study (section 3.2) and the IMAU offset (section 5.2) give an offset of the TU measurements relative to NIWA to be +0.36±0.08 ‰ (No. 6 in Fig. 2b). Measurements at TU have been regularly compared with those at NIPR and they are in agreement within reproducibility of both systems (Umezawa et al., 2009 and additional measurements since then). This is consistent with the previous intercomparison between NIPR and NIWA (section 5.7) and indicates long-term intra-laboratory consistency of TU and NIPR measurements. It is reasonable that TU shares the offset level with NIPR, because both institutions use the same standard scale. As described in section 2.6, it should be noted that the above offset value is not for the datasets currently available to the research community (Umezawa et al., 2011, 2012a, 2012b), for which +0.32±0.08 ‰ (not shown in Fig. 2) is recommended. Correction of the datasets from the earlier publications is under evaluation.

$\delta$D-$CH_4$: The intercomparison in this study (section 4.2) gives an offset of −13.1±0.6 ‰ for the TU atmospheric $\delta$D-$CH_4$ measurement (No. 6 in Fig. 2b).

## 5.7 NIPR

$\delta^{13}$C-$CH_4$: An intercomparison between NIPR and NIWA was conducted in 2004 (Morimoto et al., 2006). After the recent update of the NIPR standard scale, the NIPR offset is evaluated to be +0.33±0.04 ‰ higher than NIWA (No. 7 left in Fig. 2a). The intercomparison in this study (section 4.2)

combined with the IMAU offset (section 5.2) indicates the NIPR measurement is +0.27±0.08 ‰ with respect to NIWA (No. 7 right in Fig. 2a), consistent with the above value.

## 5.8 UW

$\delta^{13}$C-CH$_4$: Quay et al. (1999) exchanged 30 air samples with NIWA; the average measurement offset
was evaluated to be +0.02±0.14 ‰ (No. 8 left in Fig. 2a), although some individual samples disagreed by up to 0.5 ‰ (Lowe et al., 1994; Quay et al., 1999). Later, Levin et al. (2012) estimated that the UW offset is +0.058±0.004 ‰ with respect to NIWA based on co-located sampling at BHD (No. 8 right in Fig. 2a).

$\delta$D-CH$_4$: To our knowledge, no intercomparison exercises with UW have been reported.

**5.9 UHEI**

$\delta^{13}$C-CH$_4$: Levin et al. (2012) estimated the UHEI $\delta^{13}$C-CH$_4$ offset to be −0.169±0.031 ‰ relative to NIWA (No. 9 left in Fig. 2a). The intercomparison between UHEI and MPI-BGC in this study (section 3.3), together with the MPI-BGC offset (section 5.4), also infers the UHEI offset to be −0.05±0.13 ‰ (No. 9 right in Fig. 2a), consistent with the above value. Earlier measurements of three air samples at
both UHEI and NIWA indicated that the UHEI offset is −0.04±0.04 ‰ relative to NIWA (Poß, 2003; Behrens et al., 2008). It is also noted that, in an intercomparison presented by Nisbet (2005), the UHEI measurement was −0.07±0.04 ‰ lower than NIWA. As these earlier comparison results have been published before the rigorous corrections of the UHEI measurements, these values are not included in Fig. 2a.

$\delta$D-CH$_4$: The intercomparison in this study (section 4.3), combined with the MPI-BGC offset (section 5.4), indicates that UHEI has an offset of −3.8±1.3 ‰ relative to IMAU.

## 5.10 INSTAAR

$\delta^{13}$C-CH$_4$: Levin et al. (2012) estimated that the INSTAAR measurements have an offset of +0.132±0.022 ‰ with respect to NIWA (No. 10 left in Fig. 2a). In an intercomparison exercise reported
by Nisbet (2005), the INSTAAR measurement was +0.14±0.06 ‰ higher than NIWA (not shown in Fig.

2a), consistent with the above value. This study (section 4.4) indicates that the INSTAAR measurement is +0.15±0.05 ‰ higher than NIWA for the cylinder with high $CH_4$ mole fraction (CA 03560) (No. 10 middle in Fig. 2a). The intercomparison between INSTAAR and MPI-BGC (section 4.5) indicates that, combined with the MPI-BGC offset (section 5.4), the INSTAAR offset is +0.21±0.12 ‰ relative to

NIWA (No. 10 second right in Fig. 2a). Lastly, the co-located sample intercomparison (section 4.6) indicates the INSTAAR offset to be +0.08±0.11 ‰ (No. 10 right in Fig. 2a). It is important to note again that only the round robin intercomparison measurements (section 4.4 and No. 10 middle in Fig. 2a) and the intercomparison with MPI-BGC (section 4.5) were made with a PCS column to remove the Kr interference, and that the dataset currently available to the public from INSTAAR will be evaluated

for future correction.

As described in section 2.10, INSTAAR follows the standard scale of UCI. Tyler et al. (2007) reported that measurements of 10 air cylinders filled at Niwot Ridge, Colorado in 2000–2001 were analyzed at both laboratories and that the result indicated an offset of INSTAAR to be +0.04±0.12 ‰ relative to UCI. The collection of air samples at Niwot Ridge for the UCI-INSTAAR comparison continued until

2003. A revisit to the measurement record showed that the INSTAAR offset relative to UCI had shifted over the years; the average differences are +0.02±0.08 ‰ for 2000 (*N*=7), +0.12±0.07 ‰ for 2001 (*N*=2) and +0.26±0.03 ‰ for 2002 (*N*=12). This fact may suggest excursions of the internal calibration of either laboratory for these years, but the cause has yet to be resolved; this problem will be addressed in a subsequent paper from either group. It is noted that the offsets relative to NIWA for both

laboratories inferred from the different intercomparison pathways are consistent with each other within the uncertainties (Figure 2a).

$\delta$D-$CH_4$: Bock et al. (2010) reported an intercomparison between UB and INSTAAR. This indicates that the INSTAAR measurement offset is −13.2±1.3 ‰ with respect to IMAU (No. 10 in Fig. 2b).

## 5.11 RHUL

$\delta^{13}$C-$CH_4$: Nisbet (2005) reported that RHUL DI-IRMS measurements agreed well with NIWA with an offset of 0.00±0.02 ‰ (No. 11 left in Fig. 2a). At the same time, they indicated that the RHUL GC-IRMS measurement has an offset of +0.11±0.13 ‰ with respect to NIWA, and later Nisbet et al. (2016)

reported that the GC-IRMS system has an offset of about +0.3 ‰ relative to NIWA (not shown in Fig. 2a). Based on measurements of air in two cylinders exchanged between RHUL and NIWA in 2011 and 2014, RHUL applied an offset correction (−0.20 ‰) to all data (see section 4.6), by which the RHUL offset has now been evaluated to be +0.12±0.03 ‰ (No.11 middle in Fig. 2a). The intercomparisons

based on the co-located air samples via INSTAAR (section 4.6), combined with the INSTAAR offset (section 5.10), infer that the RHUL offset is +0.10±0.03 ‰ relative to NIWA (No. 11 right in Fig. 2a).

## 5.12 PDX

$\delta^{13}$C-CH$_4$: Rice et al. (2016) presented an offset of +0.024±0.088 ‰ of the PDX measurements relative to UW by comparing coinciding measurements of archive air samples at PDX and $\delta^{13}$C-CH$_4$ records

from Quay et al. (1999) from stations Mauna Loa, Hawaii and Tutuila, American Samoa (1995–1996). With the UW offset with respect to NIWA (section 5.8), it is indicated that the PDX measurement is +0.08±0.09 ‰ higher than NIWA (No. 12 in Fig. 2a). This offset is consistent with the UCI offset with respect to NIWA within the uncertainties (note that PDX follows the UCI standard scale).

$\delta$D-CH$_4$: Since PDX follows the UCI standard scale (Teama, 2013; Rice et al., 2016), the likely offset is

same as that of UCI (No. 12 in Fig. 2b).

## 5.13 PSU

$\delta^{13}$C-CH$_4$: According to Schmitt et al. (2013), the PSU measurement has an offset of +0.03±0.16 ‰ relative to NIWA after being corrected for the Kr interference. The measurements of the cylinder with high CH$_4$ mole fraction (CA 03560) at PSU are +0.03±0.16 ‰, +0.27±0.16 ‰ and +0.13±0.05 ‰ (No.

13 left, middle and right, respectively in Fig. 2a) higher than NIWA for different Kr corrections at different measurement times, these values being consistent with each other within the uncertainties.

$\delta$D-CH$_4$: An intercomparison result using three firn air samples gives the PSU offset of −12.1±1.5 ‰ relative to the IMAU measurement (Sapart et al., 2011; No. 13 left in Fig. 2b). The intercomparison in this study (section 4.4) gives −13.6±1.5 ‰ relative to IMAU for the cylinder with high CH$_4$ mole

fraction (CA 03560) (No. 13 right in Fig. 2b).

### 5.14 UB

$\delta^{13}$C-CH$_4$: The UB measurement has an offset of $-0.18\pm0.09$‰ relative to NIWA (Schmitt et al., 2013; No. 14 in Fig. 2a). This was determined by the round robin intercomparison (section 4.4 and Table 5).

$\delta$D-CH$_4$: Sapart et al. (2011) gives an intercomparison result between UB and IMAU, indicating the UB offset of $0.0\pm1.6$ ‰ relative to IMAU (No. 14 left in Fig. 2b). This value is consistent with the intercomparisons between UB and IMAU reported by Bock et al. (2010). Later UB modified the measurement set up, but the measurements of same air samples before and after all modifications were in good agreement as presented by Bock et al. (2014). The intercomparison in this study (section 3.4) shows that the UB measurement differs insignificantly by $-0.8\pm2.5$ ‰ with respect to IMAU for the cylinder with high CH$_4$ mole fraction (CA 03560) (No. 14 right in Fig. 2b).

### 5.15 AWI

$\delta^{13}$C-CH$_4$: The AWI offset is reported to be $-0.09\pm0.06$‰ with respect to NIWA (Schmitt et al., 2013; No. 15 in Fig. 2a).

### 5.16 CIC

$\delta^{13}$C-CH$_4$: Sperlich et al. (2012) reported measurements of an air cylinder at CIC, IMAU and UB. The CIC measurement insignificantly different by $+0.01\pm0.09$ ‰ from IMAU, and the CIC offset with respect to NIWA is estimated to be $-0.03\pm0.11$ (No. 16 left in Fig. 2a). They have also reported that the CIC measurement is in agreement with UB with difference of $+0.00\pm0.14$ ‰. It is noted that, although the UB offset relative to NIWA is estimated to be significant (section 5.14), the difference is still within uncertainties of the intercomparison exercises. Two pure CH$_4$ gases prepared by Sperlich et al. (2012) constitute crucial components of the reference gas series developed at MPI-BGC (Sperlich et al., 2016). This has provided a direct intercomparison between CIC and MPI-BGC. The CIC measurement is $+0.09\pm0.14$ ‰ higher than MPI-BGC. Combined with the MPI-BGC offset (section 5.4), the CIC offset with respect to NIWA is estimated to be $+0.02\pm0.18$ ‰ (No. 16 right in Fig. 2a), consistent with the aforementioned value.

$\delta$D-CH$_4$: Sperlich et al. (2016) reported $\delta$D-CH$_4$ measurement results of the two reference gases prepared by Sperlich et al. (2012) at CIC and MPI-BGC. The results indicated that the CIC measurement differs by +2.1±1.8 ‰ from MPI-BGC. Combined with the MPI-BGC offset (section 4.4), the CIC offset relative to IMAU is estimated to be −2.1±2.1 ‰ (No. 16 in Fig. 2b).

## 6 Summary and Discussion

We carried out inter-laboratory comparison exercises for atmospheric $\delta^{13}$C-CH$_4$ and $\delta$D-CH$_4$ covering many laboratories around the world. In addition, we reviewed previously published intercomparison results. The results indicated measurement offsets among laboratories, which range from −0.2 to +0.3 ‰ with respect to the NIWA DI-IRMS measurement for $\delta^{13}$C-CH$_4$ and up to −13 ‰ with respect to the IMAU measurement for $\delta$D-CH$_4$. These offset values are larger than measurement uncertainties of individual laboratories.

The significant $\delta^{13}$C-CH$_4$ measurement offsets among laboratories are obvious even though all laboratories ultimately reference to the VPDB scale. We have presented potential causes of the measurement offsets in individual laboratories (section 2), with possible further causes being hidden in all preparation and measurement steps of standard materials. (1) The scale contraction effect for DI-IRMS CO$_2$ analysis, which is instrument-dependent, could be responsible for considerable part of the observed offsets, given the fact that the atmospheric $\delta^{13}$C-CH$_4$ value (about −47 ‰) differs considerably from the primary anchor of the VPDB scale (NBS-19). (2) Individual laboratories have made calibrations against different RMs with different uncertainties of assigned values; such diverse calibration trajectories have also definitely contributed to the inter-laboratory measurement offsets. Such RMs have different chemical properties and are processed to CO$_2$ at individual laboratories, in which different fractionation is possible. (3) Different algorithms for $^{17}$O correction have been used for $\delta^{13}$C measurements at different laboratories, which could have caused biases among available datasets. (4) The Kr interference on a GC-IRMS system is in several cases a probable cause of the offsets and unfortunately, this effect is system-dependent and can vary with time, depending on the instrument settings. Lastly, it is important to note that we summarized $\delta^{13}$C-CH$_4$ measurement offsets at the

modern atmospheric $CH_4$ mole fraction level, but the offset may vary with the amount of $CH_4$ analyzed (e.g. lower mole fractions in ice core analyses, see Table 3, 5 and 6), because of a non-linear response of IRMS (Umezawa et al., 2009) and because of the Kr interference is directly dependent on the Kr to $CH_4$ ratio (Schmitt et al., 2013). Furthermore, the intercomparisons presented here focus on modern

atmospheric $CH_4$ of typically −47 ‰ and such comparisons for high and low $\delta^{13}C$-$CH_4$ values (e.g. $CH_4$ from ice cores or enriched/depleted source signatures) are to date very limited (Tables 3 and 6 in this study).

Concerning $\delta D$-$CH_4$ measurement offsets among laboratories, it is interesting that the listed laboratories can be roughly split into two groups whose $\delta D$-$CH_4$ measurements differ by ~10 ‰. Some laboratories

with higher $\delta D$-$CH_4$ values reference to an identical set of standards produced at MPIC (MPIC and IMAU) or to the UHEI calibration (UHEI and UB), and measurements of these groups have been cross-referenced (see sections 2 and 4), thereby showing the reasonable agreements. The original calibrations were made by an offline $CH_4$ processing technique (cryogenic separation and conversion of $CH_4$ to $CO_2$ and $H_2O$ followed by $H_2O$ reduction to $H_2$) with subsequent analysis by DI-IRMS. The other

laboratories with higher $\delta D$-$CH_4$ values recently developed their own primary calibrations independently (CIC and MPI-BGC). CIC used an offline $CH_4$ processing combined with DI-IRMS, whereas MPI-BGC adopted TC/EA coupled to continuous-flow IRMS. For the lower $\delta D$-$CH_4$ group, some laboratories made calibrations against Oztech $H_2$ gases (UCI, PDX and PSU) or have other calibration pathways (TU and INSTAAR) (see section 2). These laboratories used local $H_2$ working gas

standards for GC-IRMS, which were calibrated by a separate DI-IRMS procedure. As is the case for $\delta^{13}C$-$CH_4$, possible causes of the observed $\delta D$-$CH_4$ discrepancies could have arisen in all preparation and measurement steps. (1) The classical technique for DI-IRMS involves processing of $H_2O$, and the associated steps in experimental lines are prone to surface adhesion and contamination of $H_2O$, thereby considerable memory effect is possible (Bergmaschi et al., 2000). (2) Similar to $\delta^{13}C$-$CH_4$, calibration

for $\delta D$-$CH_4$ involves measurements of standards with different chemical properties ($H_2O$ and $H_2$), and such calibrations at different laboratories could contribute to the offset. (3) Difficulties in maintaining stable pyrolysis conditions for GC-IRMS (Bock et al. 2010) might have affected measurements against local $H_2$ working standards in the cases where the principle of identical treatment  (Werner and Brand,

2001) was not followed strictly. Lastly, it is noted that non-linearity of the IRMS in $\delta$D-CH$_4$ measurements (Brass and Röckmann, 2010) may also play a role for samples with low mole fractions such as ice core analyses.

The measurement offsets summarized in this study should be thoroughly taken into account when data

from different laboratories are combined, and this study will be of help when incorporating merged $\delta^{13}$C-CH$_4$ and $\delta$D-CH$_4$ datasets into a state-of-the-art chemistry transport model. However it is recommended that data users contact the data providers directly for the latest information whenever possible. The Kr interference is under evaluation at some laboratories and it will possibly involve an update of the datasets currently available. More importantly, it is imperative to have common reference

gases with transparent and reproducible traceability (for instance, Sperlich et al. 2016) and to carry out a systematic intercomparison program (flask or cylinder round robin) in the research community for attaining the necessary but ambitious high compatibility goals of 0.02 ‰ for $\delta^{13}$C-CH$_4$ and 1 ‰ for $\delta$D-CH$_4$ (WMO, 2016). Such thorough efforts will facilitate optimized use of $\delta^{13}$C-CH$_4$ and $\delta$D-CH$_4$ datasets in a combined way and maximize the number of isotope datasets (and thus their spatial and

temporal coverage) usable for enhancing our understanding of the global CH$_4$ cycle.

We welcome collaborative works to analyse the multiple datasets presented in this study (see data availability listed in Table 1). Data users can examine the offset numbers (Table 1 and Figure 2) to adjust the datasets at least for data points with values close to the modern atmosphere in $\delta^{13}$C-CH$_4$ and $\delta$D-CH$_4$ as well as CH$_4$ mole fraction. For data with CH$_4$ mole fractions and isotopic ratios that are far

from modern background values (e.g. sample air from ice core and stratosphere and those influenced by sources), more intercomparisons are needed to establish correction factors among datasets.

**List of Participating Institution/Project Acronyms**

AWI: Alfred Wegener Institute Helmholtz Centre for Polar and Marine Research, Bremerhaven, Germany

CARIBIC: Civil Aircraft for the Regular Investigation of the atmosphere Based on an Instrument Container

CIC: Center for Ice and Climate, Niels Bohr Institute, University of Copenhagen, Copenhagen, Denmark

IMAU: Institute for Marine and Atmospheric research Utrecht, Utrecht University, Utrecht, the Netherlands

INSTAAR: Institute of Arctic and Alpine Research University of Colorado Boulder, Boulder, USA

MPI-BGC: Max Planck Institute for Biogeochemistry, Jena, Germany

MPIC: Max Planck Institute for Chemistry, Mainz, Germany

NCAR: National Center for Atmospheric Research, Boulder, USA

NIPR: National Institute of Polar Research, Tokyo, Japan

NIWA: National Institute for Water and Atmospheric Research, Wellington, New Zealand

NOAA: National Oceanic and Atmospheric Administration, USA

PDX: Portland State University, Portland, USA

PSU: Pennsylvania State University, Pennsylvania, USA

RHUL: Royall Holloway, University of London, Egham, UK

TU: Tohoku University, Sendai, Japan

UB: University of Bern, Bern, Switzerland

UCI: University of California Irvine, Irvine, USA

UHEI: University of Heidelberg, Heidelberg, Germany

UW: University of Washington, Seattle, USA

**Acknowledgement**

We thank M. Sato for measurements at TU. We thank Owen Sherwood for contribution to setting up the INSTAAR measurement system, and Ken Masarie and John Mund, who developed software enabling data comparisons from co-located samples measured by INSTAAR, RHUL and NIWA. We thank Paul Quay (UW) for isotopic $CH_4$ data and intercalibration information and Doaa Teama for isotopic

measurements at PDX. We thank C. Sapart for helpful information about the past intercomparison. Part of this work at NIES was supported by the Environment Research and Technology Development Fund (2-1710) of the Ministry of the Environment, Japan and Environmental Restoration and Conservation

Agency. Funding for work at PDX was provided by the US National Science Foundation (Atmospheric and Geospace Sciences Grant 0952307). $CH_4$ isotope ice core work at UB received funding from the European Research Council under the European Union's Seventh Framework Programme (FP7/2007-2013) ERC grant agreement no 226172 and the Swiss National Science Foundation (grant no 200020_159563 & 20020_172506). NIWA's isotope measurements are funded under the Climate and Atmosphere Research Programme CAAC1804 (2017/18 SCI).

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

**Table 1. List of laboratories that conduct measurements of $\delta^{13}$C-CH$_4$ and $\delta$D-CH$_4$. For each laboratory, measurement systems and relevant information that could have contributed the inter-laboratory measurement offsets are summarized. Brackets in the RM column indicate the laboratory from which the original standard scale was propagated. See Figure 1 for overview of the past intercomparison exercises, Figure 2 for intercomparison summary and the list of participating institution/project acronyms in the text for the laboratory names.**

| No | Lab | System | IRMS | $^{17}$O correction* | Kr interference Significance | Kr interference Measure† | Additional Correction | RM $\delta^{13}$C-CH$_4$ | RM $\delta$D-CH$_4$ | Reference‡ Measurement | Reference‡ Intercomparison $\delta^{13}$C-CH$_4$ | Reference‡ Intercomparison $\delta$D-CH$_4$ | $\delta^{13}$C-CH$_4$ offset with respect to NIWA | $\delta$D-CH$_4$ offset with respect to IMAU | Data Availability§ |
|---|---|---|---|---|---|---|---|---|---|---|---|---|---|---|---|
| 1 | NIWA | DI | Nuclide (1988–1996) MAT 252 (1996–2013) MAT 253 (2013–) | C1/C2 | N | — | N | NBS-19, IAEA-CO-9, LSVEC | Not measured | R1 | R2, R3, R4, R5, R6, R7, R8, R9, R10, R11, R12, R13 | Not measured | — | Not Measured | WDCGG & NIWA website (R1, R2, R14) |
| | | CF | Isoprime | C3 | Y | DI offset/PCS | Drift correction | | | R15, R16 | | | | | On request to H. Schaefer (R17) |
| 2 | IMAU | CF | Delta plus XL | C2 | Y | PCS | Non-linearity correction for small peaks | NBS-19 (MPIC) | VSMOW & SLAP (MPIC) | R18 | R10, R12, R18, R19, R20, R21, R22, R23 | R12, R18, R20, R21, R22, R23, R24 | −0.04±0.07 | — | Utrecht University website (R25, R26) |
| 3 | MPIC | DI & TDLAS | MAT 252 | C2 | N | — | N | NBS-19 | VSMOW & SLAP | R27 | R18, R19 | R18, R24 | −0.04±0.07 | +0.3±1.1 | On request to P. Bergamaschi (R28) |
| 4 | MPI-BGC | CF | Delta V Plus | C4 | Y | PCS | Mole fraction dependent linearity correction | NBS-19 & LSVEC | VSMOW-2 & SLAP-2 | R29 | R21, R30 | R21, R30 | −0.07±0.12 | −4.2±1.2 | On request to H. Moossen (R21, R29) |
| 5 | UCI | DI | MAT 252 | C2 | N | — | N | NBS-19 & IAEA-CO-9 | Oztech Gas | R7, R31 | R7, R12, R15, R22, R32, R33 | R22, R33 | −0.01±0.09 | −10.7±0.7 | CDIAC (R31) |
| | | CF | Delta Plus XL | C2 | Y | DI offset/PCS | N | | | R7, R34 | | | | | |

| 6 | TU | CF | Delta Plus XP | C3 | Y | No | Daily correction with respect to a test gas; Constant offset for part of the dataset (R34, R35) | NBS-19 | VSMOW & SLAP | R37 | R23, R37 | R23 | +0.36±0.08 | −13.1±0.6 | WDCGG (R35) & On request to T. Umezawa (R36, R38) |
|---|---|---|---|---|---|---|---|---|---|---|---|---|---|---|---|
| 7 | NIPR | CF | MAT 252 | C3 | N | — | N | NBS-19 (TU) | Not measured | R8 | R8, R23, R37 | Not measured | +0.33±0.04 | Not measured | TU website (R8, R39) |
| 8 | UW | DI | MAT 251 | Information not available | N | — | Correction of N$_2$O produced during combustion & Drift correction | NBS-19 | VSMOW & SLAP | R3 | R2, R3, R4, R32 | No comparison available | +0.02±0.14 | No comparison | CDIAC (R3) |
| 9 | UHEI | DI | MAT 252 | C2 with coefficients a=0.5 & K=0.008335 | N | — | Drift correction of extraction and daily correction of IRMS | RM 8562, RM 8563 & RM 8564 | VSMOW & SLAP | R5 | R4, R5, R6, R19, R30 | R30 | −0.17±0.03 | −3.8±1.3 | UHEI website (R4) |
| 10 | INSTAAR | CF | Optima/Isoprime | C3 | Y | PCS | Drift correction | NBS-19 & IAEA-CO-9 (UCI) | Not Anchored | R15 | R4, R7, R9, R11, R12, R13, R15, R19, R41 | R24 | +0.13±0.02 | −13.2±1.3 | WDCGG & NOAA/ESRL/GMD website (R15) |
| 11 | RHUL | DI / CF | Prism / Isoprime | C3 | N | — | Daily offset with respect to working air standard | NBS-19 & IAEA-CO-9 | Not measured | R42 / R43 | R9, R11, R13 | Not measured | +0.12±0.03 | Not measured | CEDA & On request to E. Nisbet (R11) |
| 12 | PDX | CF | Delta V | C2 | Y | DI offset/PCS | N | NBS-19 & IAEA-CO-9 (UCI) | Oztech Gas (UCI) | R33, R34 | R32, R33 | R32, R33 | +0.08±0.09 | −10.7±1.5 | PDX website (R32) |
| 13 | PSU | CF | MAT 252 | C2 | Y | Raw ion current correction/ DI offset/PCS | Daily offset with respect to primary air standard | RM 8563 & RM 8564 | Oztech Gas | R20, R44 | R10, R12, R19, R20 | R12, R20, R24 | +0.03±0.16 | −12.1±1.5 | NSIDC & On request to T. Sowers |
| 14 | UB | CF | Isoprime | C2 | Y | PCS | Drift correction | NBS-19 & IAEA-CO-9 (UCI via | VSMOW & SLAP (UHEI) | R10, R24, R41, | R10, R12, R20, | R12, R20, R24, | −0.18±0.09 | 0.0±1.6 | PANGAEA (R46, R47) |

| No | Lab | Method | Instrument | Std* | | Kr† | Correction | RM | | | | | | | Data |
|---|---|---|---|---|---|---|---|---|---|---|---|---|---|---|---|
| | | | | | | | | INSTAAR) | | R45 | R41 | R45 | | | |
| 15 | AWI | CF | Isoprime | C3 | Y | No | Drift correction | RM 8562, RM 8563 and RM 8564 (UHEI) | Not measured | R6 | R6, R10, R19, R20 | Not measured | −0.09±0.06 | Not measured | PANGAEA (R19) |
| 16 | CIC | DI ($\delta^{13}$C-CH$_4$) TC/EA-IRMS & Picarro ($\delta$D-CH$_4$) | Delta V Plus, Delta V Advantage and Picarro | C2 | N | — | — | RM 8563 | VSMOW-2 & SLAP-2 | R48 | R21, R48 | R21, R48 | −0.03±0.11 | −2.1±2.1 | All data in papers (R16, R21, R48) |
| | | CF | Delta V Plus | C2 | Y | PCS | Daily offset with respect to working gas standard; CH$_4$ amount correction | | | Not measured | R16 | R16 | | | |

*C1: Allison et al. (1995), C2: Santrock et al. (1985), C3: Craig (1957), C4: Assonov and Brenninkmeijer (2003)

†Raw ion current correction: The Kr interference was corrected by subtracting the Kr-caused anomalies in the raw ion current data; DI offset: The Kr interference was corrected by an offset relative to a DI-IRMS measurement; PCS: Kr was separated by a post combustion separation column or cryogenically. See section 2.5.

‡R1: Lowe et al. (1991), R2: Lowe et al. (1994), R3: Quay et al. (1999), R4: Levin et al. (2012), R5: Poß (2003), R6: Behrens et al. (2008), R7: Tyler et al. (2007), R8: Morimoto et al. (2006), R9: Nisbet (2005), R10: Schmitt et al. (2013), R11: Nisbet et al. (2016), R12: This study (Section 4.4), R13: This study (Section 4.6), R14: Bergamaschi et al. (2001), R15: Miller et al. (2002), R16: Sperlich et al. (2013), R17: Ferretti et al. (2005), R18: Brass and Röckmann (2010), R19: Möller et al. (2013), R20: Sapart et al. (2011), R21: Sperlich et al. (2016), R22: This study (Section 4.1), R23: This study (Section 4.2), R24: Bock et al. (2010a), R25: Röckmann et al. (2010), R26: Röckmann et al. (2016), R27: Bergamaschi et al. (1994), R28: Bergamaschi et al. (2000); R29: Brand et al. (2016), R30: This study (Section 4.3), R31: Tyler et al. (1999), R32: Rice et al. (2016), R33: Teama (2013), R34: Rice et al. (2001), R35: Umezawa et al. (2012a), R36: Umezawa et al. (2012b), R37: Umezawa et al. (2009), R38: Umezawa et al. (2011), R39: Morimoto et al. (2017), R40: This study (Section 4.3), R41: Schmitt et al. (2014), R42: Lowry et al. (2001), R43: Fisher et al. (2006), R44: Sowers et al. (2005), R45: Bock et al. (2014); R46: Bock et al. (2010b), R47: Bock et al. (2017), R48: Sperlich et al. (2012)

§WDCGG (World Data Centre for Greenhouse Gases): http://ds.data.jma.go.jp/gmd/wdcgg/wdcgg.html
NIWA website: www.niwa.co.nz
Utrecht University website: http://www.projects.science.uu.nl/atmosphereclimate/Data.php
TU website (http://caos.sakura.ne.jp/tgr/data/en)
CDIAC (Carbon Dioxide Information Analysis Center): http://cdiac.ess-dive.lbl.gov/epubs/db/db1022/db1022.html (UCI), http://cdiac.ess-dive.lbl.gov/ndps/quay.html (UW)
UHEI website: www.iup.uni-heidelberg.de/institut/forschung/groups/kk/Data_html
NOAA/ESRL/GMD website: https://www.esrl.noaa.gov/gmd/dv/data/
PDX website: http://web.pdx.edu/~arice/atm_CH4.html
PANGAEA: https://doi.org/10.1594/PANGAEA.873918
CEDA (Centre for Environmental Data Analysis): http://www.ceda.ac.uk
H. Schaefer: h.schaefer@niwa.co.nz
P. Bergamaschi: peter.bergamaschi@ec.europa.eu
H. Moossen: heiko.moossen@bgc-jena.mpg.de
T. Umezawa: umezawa.taku@nies.go.jp
E. G. Nisbet: e.nisbet@rhul.ac.uk
T. Sowers: tas11@psu.edu

**Table 2. Result of intercomparison of $\delta^{13}$C-CH$_4$ and $\delta$D-CH$_4$ measurements between UCI and IMAU.**

| Sample ID | CH$_4$ (ppb) [*] | $\delta^{13}$C-CH$_4$ (‰) [†] UCI | IMAU | UCI−IMAU | $\delta$D-CH$_4$ (‰) [†] UCI | IMAU | UCI−IMAU |
|---|---|---|---|---|---|---|---|
| WAS-24-2 | 1784.7 | −46.96±0.07 (*N*=3) | −47.33±0.05 (*N*=3) | +0.37 | −91.6 (1.0, *N*=2) | −78.9±0.1 (*N*=4) | −12.7 |
| WAS-24-5 | 1825.8 | −47.16 (*N*=1) | −47.53±0.02 (*N*=6) | +0.37 | −93.8 (*N*=1) | −83.1±0.2 (*N*=4) | −10.7 |
| WAS-24-6 | 1827.5 | −47.08 (0.02, *N*=2) | −47.55±0.04 (*N*=6) | +0.47 | −92.1 (1.6, *N*=2) | −83.6±0.1 (*N*=4) | −8.5 |
| WAS-24-9 | 1799.8 | −47.05 (*N*=1) | −47.38±0.02 (*N*=6) | +0.33 | −92.3±1.8 (*N*=3) | −79.8±0.8 (*N*=4) | −12.4 |
| WAS-24-10 | 1789.8 | −47.07 (*N*=1) | −47.42±0.02 (*N*=6) | +0.35 | −89.3 (*N*=1) | −79.7±0.8 (*N*=4)) | −9.6 |
| WAS-24-11 | 1780.8 | −46.77 (*N*=1) | −47.37±0.03 (*N*=6) | +0.60 | −89.0 (1.8, *N*=2) | −78.7±0.7 (*N*=4) | −10.3 |
| **Average** | | | | +0.42±0.04[‡] | | | −10.7±0.7[‡] |

[*]NOAA-2004 CH$_4$ scale (Dlugokencky et al., 2005)

[†]Uncertainties are standard errors of the mean for measurements with $N \geqq 3$. Difference of duplicate flask measurements (*N*=2) is shown in parenthesis.

5    [‡]Uncertainties are standard errors of the mean for differences in the above lines.

**Table 3. Result of intercomparison of $\delta^{13}$C-CH$_4$ and $\delta$D-CH$_4$ measurements between TU/NIPR and IMAU.**

| Sample ID | CH$_4$ (ppb)[*] | $\delta^{13}$C-CH$_4$ (‰) [†] | | | | $\delta$D-CH$_4$ (‰) [†] | | |
|---|---|---|---|---|---|---|---|---|
| | | TU | NIPR | IMAU | Difference from IMAU | TU | IMAU | Difference from IMAU |
| MD1 | 1901.1 | −47.04±0.02 (N=16) | −47.11±0.02 (N=5) | −47.40±0.04 (N=9) | +0.36 (TU) +0.28 (NIPR) | −97.2±0.6 (N=10) | −85.0±0.1 (N=8) | −12.2 |
| MD2 | 2116.6 | −46.81±0.02 (N=16) | −46.92±0.03 (N=6) | −47.26±0.03 (N=9) | +0.45 (TU) +0.34 (NIPR) | −118.5±0.6 (N=10) | −104.5±0.3 (N=8) | −14.0 |
| MD3 | 899.1 | −41.14±0.04 (N=16) | −41.05±0.02 (N=5) | −41.81±0.03 (N=8) | +0.67 (TU) +0.76 (NIPR) | −190.7±0.6 (N=10) | −175.8±0.6 (N=8) | −14.9 |
| MD4 | 1700.5 | −42.47±0.03 (N=16) | −42.43±0.04 (N=5) | −42.98±0.02 (N=8) | +0.52 (TU) +0.56 (NIPR) | −195.2±0.6 (N=10) | −180.6±0.2 (N=8) | −14.6 |
| **Average (ambient air)** | | | | | +0.40±0.04 (TU) [‡] +0.31±0.03 (NIPR) [‡] | | | −13.1±0.6 [‡] |
| Average (all) | | | | | +0.50±0.07 (TU) [‡] +0.48±0.11 (NIPR) [‡] | | | −13.9±0.9 [‡] |

[*]Tohoku University CH$_4$ scale (Aoki et al., 1992; Umezawa et al., 2014)

[†]Uncertainties are standard errors of the mean for the repetitive measurements.

[‡]Uncertainties are standard errors of the mean for differences in the above lines.

**Table 4. Result of intercomparison of $\delta^{13}$C-CH$_4$ and $\delta$D-CH$_4$ measurements between UHEI and MPI-BGC.**

| Sample ID (Collection Date) | Preparation Date UHEI | Analysis Date MPI-BGC | $\delta^{13}$C-CH$_4$ (‰) | | | $\delta$D-CH$_4$ (‰) | | |
|---|---|---|---|---|---|---|---|---|
| | | | UHEI | MPI-BGC[*] | UHEI −MPI-BGC | UHEI[*] | MPI-BGC[*] | UHEI −MPI-BGC |
| GvN 88/20 (24 Jul. 1988) | 17 Dec. 2003 | 9 Jul. 2013 | −47.54 (N=1) | −47.66 (0.07, N=2) | +0.13 | −83.3 (N=1) | −82.1 (0.8, N=2) | −1.2 |
| GvN 92/12 (11 May. 1992) | 11 Dec. 2008 | 17 Jun. 2013 | −47.43 (N=1) | −47.40 (0.04, N=2) | −0.03 | −79.1 (N=1) | −81.2 (0.9, N=2) | +2.1 |
| GvN 96/03 (13 Feb. 1996) | 11 Nov. 2003 | 17 Jun. 2013 | −47.27 (N=1) | −47.18 (0.26, N=2) | −0.08 | −73.9 (N=2) | −74.6 (0.9, N=2) | +0.8 |
| GvN 99/14 (29 Dec. 1999) | 3 Apr. 2003 | 9 Jul. 2013 | −47.30 (N=1) | −47.23 (0.16, N=2) | −0.07 | −75.2 (N=2) | −74.6 (1.3, N=2) | −0.5 |
| GvN 06/14 (23 Sep. 2006) | 7 May. 2003 | 9 Jul. 2013 | n.a. | −47.19 (0.09, N=2) | n.a. | −72.3 (N=1) | −73.1 (0.0, N=2) | +0.8 |
| GvN 08/03 (6 Mar. 2008) | 28 Jul. 2010 | 17 Jun. 2013 | −47.18 (N=1) | −47.35 (0.05, N=2) | +0.17 | n.a. | −67.4 (2.9, N=2) | n.a. |
| **Average** | | | | | +0.02±0.05[†] | | | +0.4±0.6[†] |

[*]Difference of duplicate flask measurements is shown in parenthesis.

[†]Uncertainties are standard errors of the mean for differences in the above lines.

**Table 5. Results from the Ice Core Intercomparison Round Robin conducted during 2007–2016.**

| Laboratory | CA 03560 (1830.6 ppb) | | CA 71560 (904.0 ppb) | | CA 01179 (372.2 ppb) | | Kr corr. | Analysis Date $\delta^{13}$C-CH$_4$ | Analysis Date $\delta$D-CH$_4$ |
|---|---|---|---|---|---|---|---|---|---|
| | $\delta^{13}$C-CH$_4$ (‰)[†] | $\delta$D-CH$_4$ (‰)[†] | $\delta^{13}$C-CH$_4$ (‰)[†] | $\delta$D-CH$_4$ (‰)[†] | $\delta^{13}$C-CH$_4$ (‰)[†] | $\delta$D-CH$_4$ (‰)[†] | | | |
| PSU | −47.20±0.16 | −93.2±0.9 | −47.41±0.10 | −95.5±2.3 | −47.52±0.06 | −106.3±2.4 | Raw ion current circrection[a] | Jul. 2007 | Jul. 2007 |
| | −46.96±0.16 | | −47.20±0.10 | | −47.41±0.12 | | DI offset[b] | Jul. 2007 | |
| | −47.10±0.05 | | −47.09±0.06 | | −46.83±0.12 | | PCS[c] | May 2016 | |
| UCI (DI-IRMS) | −47.09±0.12 | | −47.40±0.08 | | −47.23±0.06 | | — | Dec. 2007[*] | |
| INSTAAR | −47.08±0.05 | | −47.20±0.06 | | −46.78±0.06 | | PCS[c] | Dec. 2008 | |
| NIWA (DI-IRMS) | −47.23±0.02 | | −47.44±0.02 | | −47.43±0.02 | | — | Jun. 2009 | |
| NIWA (GC-IRMS) | −47.44±0.21 | | −48.34±0.28 | | −47.62±0.11 | | DI offset[b] | Jun. 2009 | |
| UB | −47.41±0.09 | −80.4±2.2 | −47.37±0.07 | −81.0±2.0 | −47.31±0.11 | −86.2±3.3 | No[d] | Jan. 2011 | Dec. 2010–Jan. 2011 |
| IMAU | −47.27±0.07 | −79.6±1.2 | −47.52±0.11 | −83.6±3.8 | −47.20±0.20 | −78.8±12.4 | PCS[c] | May & Aug. 2012 | May 2010 |

[a]Raw ion current correction: The Kr interference was corrected by subtracting the Kr-caused anomalies in the raw ion current data (section 5.4 of Schmitt et al., 2013); [b]DI offset.: The Kr interference was corrected by an offset relative to a DI-IRMS measurement; [c]PCS: Kr was separated by a post combustion separation column (section 5.2 of Schmitt et al. 2013); [d]No: Measurements are affected by the Kr interference (old system without PCS) and raw ion current correction was not possible.

[*]Estimated because no exact record on the analysis date at UCI is unfortunately available.

[†]Uncertanties are standard deviations of multiple measurements at each laboratory.

**Table 6. Results of $\delta^{13}$C-CH$_4$ intercomparison between INSTAAR and MPI-BGC.**

| Sample ID | CH$_4$ (ppb)[*] | $\delta^{13}$C-CH$_4$ (‰) [†] INSTAAR | MPI-BGC | INSTAAR−MPI-BGC |
|---|---|---|---|---|
| HUEY-001 | 1905.5 | −47.37±0.01 | −47.67±0.01 | +0.29 |
| | | (N=22) | (N=24) | |
| DEWY-001 | 1879.9 | −47.38±0.01 | −47.67±0.01 | +0.28 |
| | | (N=26) | (N=22) | |
| LOUI-001 | 1496.0 | −47.26±0.01 | −47.55±0.02 | +0.29 |
| | | (N=17) | (N=22) | |
| CART-001 | 1848.1 | −42.98±0.01 | −43.30±0.03 | +0.32 |
| | | (N=21) | (N=7) | |
| STAN-001 | 1696.4 | −56.60±0.01 | −57.20±0.05 | +0.60 |
| | | (N=7) | (N=8) | |
| KENN-001 | 1847.6 | −47.65±0.01 | −47.94±0.05 | +0.28 |
| | | (N=26) | (N=7) | |
| KYLE-001 | 1847.6 | −47.27±0.01 | −47.51±0.07 | +0.24 |
| | | (N=29) | (N=6) | |

[*]NOAA-2004 CH$_4$ scale (Dlugokencky et al., 2005)

[†]Uncertainties are standard errors of the mean for the repetitive measurements.

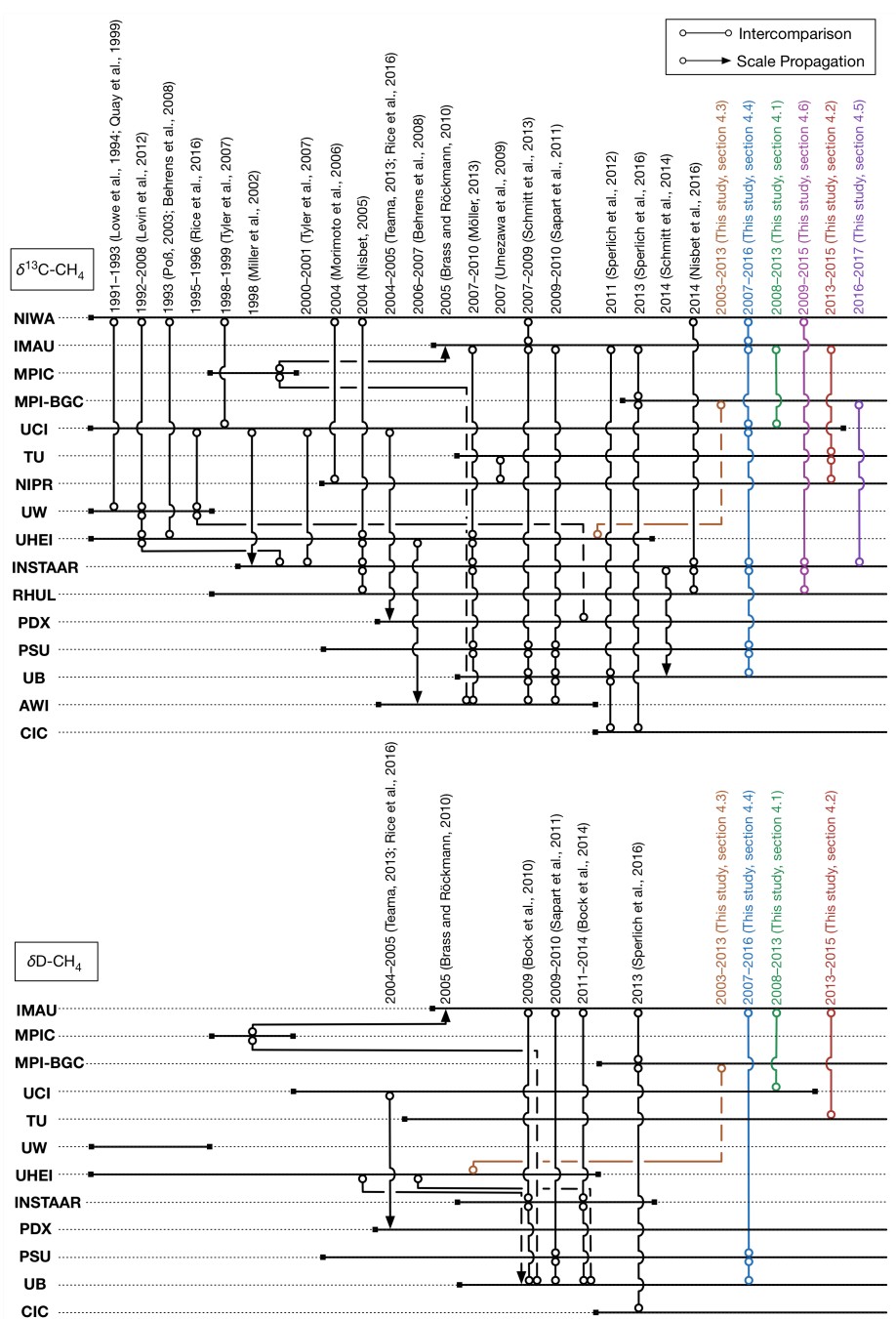

**Figure 1. A schematic overview of the previous and present intercomparisons among laboratories for $\delta^{13}$C-CH$_4$ (top) and $\delta$D-CH$_4$ (bottom). Intercomparisons are marked by lines with open circles at both ends, and scale propagations are by lines with an arrow at one end.**

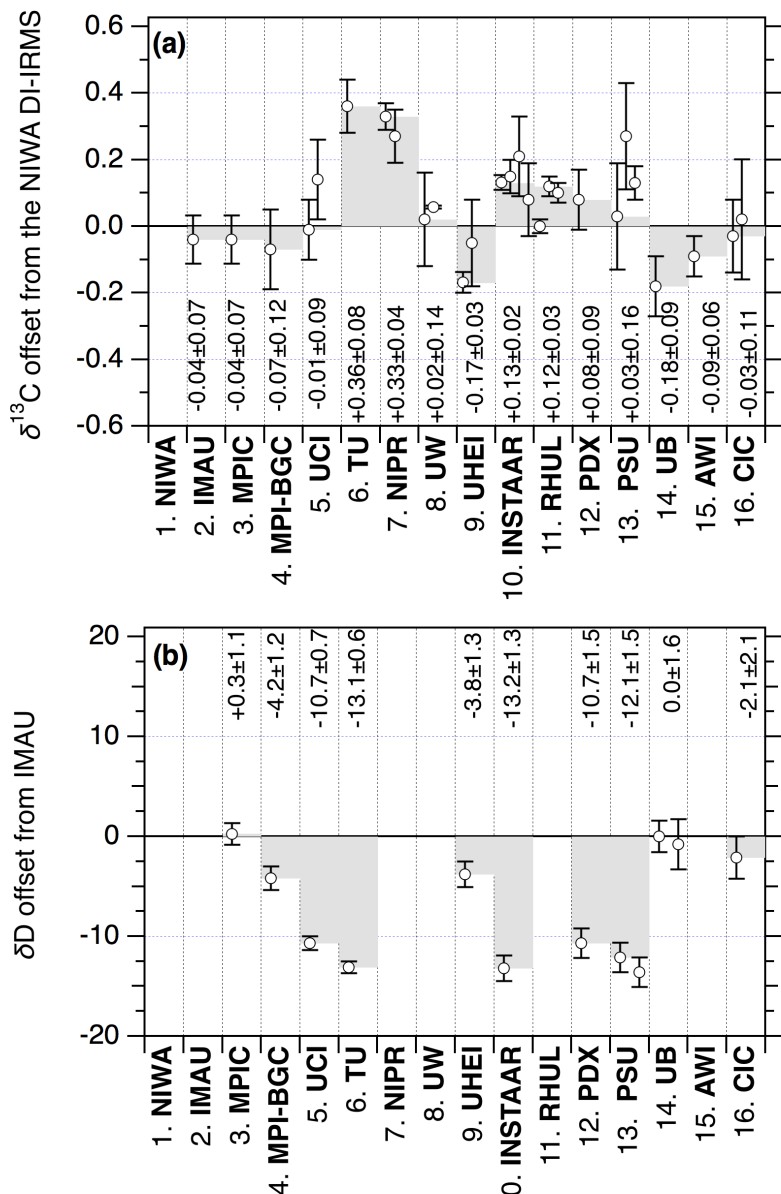

**Figure 2. (a)** $\delta^{13}$C-CH$_4$ offsets of the different laboratories with respect to the NIWA DI-IRMS measurement with gray shades for ease of viewing. **(b)** $\delta$D-CH$_4$ offsets of the different laboratories with respect to the IMAU GC-IRMS measurement. See Table 1 and text for corresponding subsections in sections 3 and 5. Numbers shown in each laboratory column are the plausible measurement offsets estimated in this study. Note that this result represents intercomparisons for the modern atmospheric CH$_4$.