# Peer review of "Inter-laboratory comparison of $\delta^{13}C$ and $\delta D$ measurements of atmospheric $CH_4$ for combined use of datasets from different laboratories"

_Atmospheric Measurement Techniques, 2017_

## Short Comment (SC1) · 17 Oct 2017

Review comments and suggestions are given in the PDF attached.

Please also note the supplement to this comment:
https://www.atmos-meas-tech-discuss.net/amt-2017-281/amt-2017-281-SC1-supplement.pdf

---

## Referee Comment (RC1) · S.S. Assonov (Referee) · 19 Oct 2017

Stable isotopes of methane, an important greenhouse gas, are used to understand the budget of methane sources and atmosphere oxidation processes. In particular, isotopes are needed to understand why the growth of atmospheric methane was levelled off in 2006-2007 and later renewed (Nisbet et al., 2016). From 2006 to 2016, δ13C(CH4) is discussed to be shifted to negative direction for ~0.3 ‰ (Nisbet et al., 2016). In order to interpret the isotope data, first of all one has to obtain reliable, high precision and high accuracy data, to be based on reliable calibrations. The problem is that offsets between labs are larger than the measurement reproducibility of each individual lab, in particular the intercomparison Round Robin in 2007–2016 demonstrated discrepancies up to ~0.6 ‰ in δ13C and ~36 ‰ in δ2H (Table 5). In the manuscript, the authors review the situation with calibrations, inter-comparisons and lab-to-lab discrepancies.

All in all, the authors have performed a great work to make a summary of numerous measurements done in several labs, their calibration histories and demonstrated discrepancies between data sets produced. However, it is evident that reliable data sets and even data synchronisation cannot be obtained at the moment, and much more work is needed.  The authors are in the position to make analysis of the situation and make suggestions on further steps necessary to obtain reliable data in the future; all that has to be listed in the abstract and stressed in conclusions. Suggestions and/or recommendations on calibration practices, use of reference materials and design of inter-comparisons would be valuable both for laboratories and also for Global Atmospheric Watch program at WMO. Biannual WMO-IAEA meetings on CO2 and greenhouse gas measurement techniques are focused on improvements needed, including stable isotope measurements of atmospheric CO2 and methane (e.g. GGMT-2015, GAW Report No 229).

The authors express the hope that CH4-mixtures (Sperlich et al., 2016) to be made in the way similar to CO2-in-air calibration mixtures being produced by MPI-BG (JRAS-mixtures) may be of great help. However, GGMT-2015 has recognised that after introducing JRAS-mixtures, lab-to-lab discrepancies in δ13C(air-CO2) demonstrated by intercomparisons have not be decreased (GAW Report No 229) and much work is still needed. Thus, careful analysis of the current situation with methane isotopes as well as focused recommendations need to be give. If the authors are not in a position to come to concise recommendations, at least they should give a better summary of their work and analysis.

In general, the manuscript is extremely long, not easy to follow and understand causes of the problems. It can be further optimised by grouping some problems and then addressing these groups. Given that mostly δ13C has been analysed, the reviewer focuses on δ13C data, the same aspects are mostly valid for d2H data.

First, the reviewer suggests grouping problems related to calibrations as following (some aspects are addressed below in more details):

1. Instrumental effects and raw data corrections. These include (i) cross-contamination (memory) in the mass-spectrometer ion sources; this effect shrinks the δ13C values, namely the distance between NBS19-CO2 used for many calibrations (δ13C of 1.95 ‰) and δ13C values of samples being around -47 ‰; (ii) consistent use of 17O correction for raw CO2 mass-spectrometry data, (iii) Kr-effect in continuous flow mode of mass-spectrometry runs (before optimisations) and its magnitude. Notably, the aspects (i)-(ii) are not addressed in the manuscript.

2. Consisted use of Reference Materials (RMs) and data management. Isotope trends can be understood in a reliable way if, and only if data are correctly positioned on the scale. Absence of any drifts can be demonstrated by re-calibrations vs. reliable RMs, to be repeated on regular basis. (The reviewer has not found much information on repeated calibrations.) First, one has to pay attention to the fact that δ13C values of RMs have been revised in the past (e.g. Coplen et al., 2006) and updated values have to be taken. This implies the need for data management & data archiving in the way allowing data reprocessing retroactively. Second, there is no reliable RMs aimed at δ13C in methane, and such dedicated RM(s) is urgently needed. In

particular, LSVEC, the high-level RM aimed at δ13C=-46.60 ‰ introduced for data normalisation by (Coplen et al., 2006) is found to be unstable. The scatter in δ13C observed on different LSVEC aliquots is ~0.35 ‰, with the LSVEC value drifting in time due to adsorption of air-CO2 (Assonov et al. in GGMT-2015, GAW Report 229). Calibrations based on NBS19-CO2 & NBS18-CO2 may be biased due to cross-contamination effect (see above) and dedicated instrumental tests have to be performed.

3. Practical calibration approaches and inconsistencies. Careful calibrations and regular re-calibrations is a must. Contrary, a common practice is to transfer calibrations from one lab to another (e.g. by transferring characterised CO2 or CH4 gas, or by performing calibration measurements for another lab, several examples are listed below). This practice cannot be recommended as it may, and often does bring to unpredictable biases (e.g. due to inconsistent use of the 17O correction) and also precludes correct data management (corrections to be applied in a consistent way and/or corrections to be applied retrospectively). This and the fact that revision of δ13C values for RMs took place in 2006 (Coplen et al., 2006) , this demands for a careful revision of calibrations in each lab.

4. Routine protocols and operating procedures have to be established in each lab and followed, both for calibrations (more critical) and for sample analysis. Any deviation from established protocols (e.g. due to rotation of personnel) may bring to unrecognised bias.

In order better summarise all the aspects related to calibrations and make analysis, the reviewer suggests making a large table, listing all the labs and providing columns corresponding to each problem and how it has been addressed. This may be helpful to visualise the situation, adopted calibration approaches and inconsistencies. Without a careful analysis and guidance, numerous examples given in the manuscript may result in some misunderstanding and misleading.

Second, there is a common misbelief about inter-comparison activities and round robins. These cannot replace factual calibrations (and regular re-calibrations) and also cannot help bringing data on the VPDB δ13C scale in reliable way. The inter-comparisons (e.g. round robin conducted during 2007–2016 demonstrate large discrepancies, up to 0.6 ‰ in δ13C and 36 ‰ in δ 2H (Table 5) and indicate problems only. As one cannot exclude calibration's biases and/or drifts, non-stable instrumental effects, change in lab 'operation procedures and other problems, lab-to-lab discrepancies in may differ over years. Systematic understanding of inter-comparison results is hardly possible without dedicated design of inter-comparisons; this has to include synchronised use of RMs, synchronised corrections for instrumental effects and synchronised 17O correction. In particular, inter-comparisons have to be done on the same material (same samples or sample archives, or artificial mixtures) and preferably addressing 1-2 effects (e.g. Kr-related baseline, calibration bias).

Third, data management and uncertainty propagation. Similar to air-CO2 isotope data, archiving all raw mass-spectrometry data as well as details of all calibrations can be recommended (e.g. see GAW-229 Report, GGMT-2015). Then uncertainties can be propagated, in order the uncertainty budget to be used as a tool aimed to demonstrate critical steps and improvements needed. However, the messy situation with calibrations, data management, corrections etc, all that precludes correct uncertainty propagation scheme.

Below some more details are given, starting with the problems related to calibrations:

1. Cross-contamination (memory) in the mass-spectrometer ion source is known to shrink the δ13C distance between sample gas and mass-spectrometer reference CO2 gas. The magnitude of cross-contamination effect on MAT252 is reported to be up to a few 0.1 ‰ (Verkouteren et al., 2003a, 2003b). In particular this is relevant to pre-2003 works (e.g. δ13C measurements at MPI-C are done before 2003) when no specific measures have been taken such as Tantalum slits in the ion source and optimisation of ion source tuning.

2. All δ13C data are obtained by correcting raw $CO_2$ mass-spectrometry data for 17O contribution, this is so-called the 17O correction. The 17O correction after Craig (1957) modified by Allison et al. (1995) and the one after Santrock et al (1985) has been used for many years; later the 17O correction was re-determined by Assonov & Brenninkmeijer (2003) and this correction has been recommended by IUPAC as the most accurate one and avoiding biases in δ13C (Coplen et al., 2006; Brand et al., 2010). Inconsistent use of the 17O corrections (e.g. by transfer of calibrations from one lab to another) and/or comparing data sets obtained in different years and using different 17O corrections may bring to unpredictable bias(es). Notably, revision of δ13C values for RMs (Coplen et al., 2006) is partly related to the use of 17O-correction after Assonov & Brenninkmeijer (2003).

3. One needs to stress that Kr affects the continuous flow mode of mass-spectrometry only; one may give an estimate of the magnitude and its direction.

The reviewer suggests listing all the cases of calibration' transfer from one lab to another one, again this may be given in a table. Several examples of calibration transfer (not all cases) are listed below:

- from MPIC to IMAU (page 6, lines 5-8);
- from IMAU to MPI-BC (p. 7, l. 5-6), I cite "Initially, the GC-IRMS measurements had been anchored to a working standard air calibrated by IMAU." Note, IMAU has transferred calibration from MPIC, see above.
- from Bundesanstalt für Geowissenschaften und Rohstoffe to MPIC (p. 6, l. 22-23), I cite: "The MPIC δD-CH4 scale is based on measurements of standard gases at the Bundesanstalt für Geowissenschaften und Rohstoffe, Hannover, Germany";
- from UEHI to AWI, I cite: "The δ13C-CH4 measurements follow the UHEI calibration via comparison of measurements of an Antarctic air sample (Möller et al., 2013)."
- from INSTAAR to UB, I cite: "δ13C-CH4 value of −47.34±0.02 ‰ ……………… is anchored to the INSTAAR calibration"

There are examples of incorrect use of δ13C values for RMs.

- For UCI' calibration, reference is given to (Rici et al., 2001), and here I cite from Rici : "values of 1.92 and -47.18‰ for NBS-19 and IAEA-CO-9, respectively." There are 2 problems, namely NBS19 has by definition the value of 1.95 ‰ (no revision followed) whereas the latest revision of δ13C values (Coplen at al., 2006) gave for IAEA-CO-9 the value of -47.32 ‰; the difference of 0.14 ‰ is not negligible.
- For TU, Umezawa et al. (2009) takes NBS18' recommended δ13C value as 5:029 ‰ whereas the latest revision by Coplen et al. (2006) gives for value of -5.01 ‰ (the difference 0.02 ‰).

There may be cases of unexplained drifts and effects of large magnitude, such as one reported for TU. The manuscript refers to Umezawa et al. (2012) and, I cite from this work: "The measured 13C value of the test gas was stable to be (−47.12±0.10) ‰ from the start date of our measurement until April 2008. Afterward, the measured value suddenly shifted to (−46.85±0.09) ‰, keeping the same precision as before. The cause is still unclear, since we had not changed any measurement settings. To keep data consistency, we added −0.27‰ to the measured values after the gap."

S.Assonov,

17-10-2017

---

## Referee Comment (RC2) · Anonymous Referee #2 · 26 Oct 2017

The authors take on the difficult and important task of summarizing comparisons of measurements of CH4 stable isotopes in air at CH4 levels appropriate for the current atmosphere and air extracted from ice cores. Done properly, this would give data users correction factors to combine data from different laboratories and give them an understanding of the major issues involved so they fully understand the limitations of the combined data sets. This would allow more CH4 isotope data to be used in studies of the global CH4 budget. Unfortunately, the paper seems to be written for isotope measurement experts, like the authors, rather than for data users who may be very interested in more CH4 isotopic composition measurements in their studies. It contains too much jargon and too little explanation of the major issues that prevent labs

from preparing a combined data set for CH4 isotopes with meaningful temporal and spatial gradients. These issues seem to be inherent in the the community's measurement approach. But what are those issues? Are they associated with deficiencies in measurement techniques themselves and how instruments are calibrated? Is there a lack of appropriate isotope standards for CH4? Is the issue with propagating standard scales from carbonates (13C) or water (D/H) to CH4 in air? The fundamental hierarchy of standards used within the CH4 isotope community should be described; much of what is described seems to violate good metrological practice. Are sample collection or processing methods causing differing amounts of fractionation among labs? These are just some possibilities from a non-expert. I suggest that a brief, systematic description of the important issues involved in making measurements of atmospheric CH4 isotopic composition and ultimately preparing a consistent data set across measurement labs is given in Section 2. This would be especially useful to data users and also help the authors focus on how to move forward.

This study is supposed to help scientists utilize more CH4 isotopic data by providing correction factors to make data from different labs more compatible, but are the data sets available? A quick look at the World Data Center shows only NIWA and NOAA data have been updated within the past couple years for 13CH4, and the only other data set, quite outdated, is from Tohoku University. Are the data from ice core and firn air available? Do other labs make data available through their institutes' web sites? That issue aside, using the comparison information in the paper is complicated by Kr interference; it is not always clear where a correction has been applied and, as a result, what data sets the information in this paper is appropriate for. It gives the impression that the paper was written before the isotope measurement community is ready for such an effort to be more generally useful to the CH4 research community.

It was surprising that XCH4 (i.e., CH4 mole fractions) was not reported in the paper, at least for the cylinders that were circulated as part of this study. It is important that CH4 measurements agree among labs measuring air from ice cores and the modern

atmosphere to get radiative forcing correct, and this is a good forum to show that level of agreement.

The manuscript is poorly written. It is far too wordy. While the first author is not a native English speaker, there are at least 10 authors who are. Shame on all of you for not improving the English. Language is often vague. Scientific terms are misused.

Overall, I think the work described in the paper is important, despite that the community is not yet capable of preparing useful combined data sets. I recommend that the paper is re-evaluated for acceptance after the authors respond to the comments in this review to the satisfaction of the journal editor.

General comments: 1. Use appropriate metrological terms. "Precision" is a qualitative term, yet it is used as a quantitative measure of the noise or uncertainty in a measurement system. Is it short-term noise (repeatability) or does it represent long-term variations of a measurement system (reproducibility)? When the proper terms are used, how are they quantified? More appropriately, uncertainty should be stated with its confidence interval.

2. Calibration: paraphrased, it links the measured response of an analyzer to the known values of standards. In the text, it seems to be confused with standard, and its use is often unclear. Given that, terms like "calibration offsets" are vague. Are the standards different? Is the issue with propagation of the standards? Could the offset result from fractionation during sample processing?

3. Differences between labs are often given with standard errors; I think standard deviation would be a better metric. In cases where n is large, e.g., for ongoing comparisons that happen over years, standard error exaggerates how well the difference is known.

4. Remove unnecessary words: assessed to be... comparison exercises (delete "exercises") evaluated to be X L in volume (delete "in volume") considered to be "in the time period" and "the years" offset value (delete "value")

[Figure]

5. Each participant in measurement of CH4 isotopes from the circulated cylinder should report XCH4.

6. "Concentration" is misused. In most cases, it can be deleted, because the unit provided (ppb) defines the measured quantity.

7. Why were other labs measuring CH4 isotopic composition not included (e.g., Oregon State University)?

8. What are the main sources of differences among labs? You imply it is differences in standard scales, but why? Don't the working standard scales propagate back to a primary scale, e.g., VPDB for 13C?

9. What is the path forward, beyond what was mentioned regarding newly-developed standards? Many deviations from good metrological practice, especially regarding propagation of isotope standard scales, have occurred within this community. The new standards, although developed using an approach that may not defensible in a pure metrological sense, seems practical given the limited resources of the measure­ment community. Is that alone sufficient? What else needs to be done to make existing data more compatible? How could new laser-based spectroscopic methods help this measurement community and the science? What else could improve the quality and compatibility of measurements of CH4 isotopic composition across the labs involved here and beyond to others? As mentioned, data availability is not considered.

Specific comments:

P1L32: suggest ..from an inter-laboratory comparison of measurements.... (Also for title.)

P1L32: ..among worldwide...

P2L3: What does "the data" refer to? The differences among labs?

P2L4-5: As presented, it is not clear how this will help combine data sets. It could

be more clear it a table was given of available data sets and if (i.e., with respect to Kr interference), and how, the offsets in the paper apply.

P2L8-12: The description of how CH4 isotopes are used to constrain the CH4 budget could be stated more clearly. I suggest something like "The mass-weighted average delta-13C of emissions from all sources will equal the delta-13C of atmospheric CH4, after correction for fraction by removal processes." While you give some references for studies that use isotopes of CH4 (some are poor examples), you don't reference early literature that identified their usefulness (e.g., Stevens).

P2L21: Is not this ratio more generally rare/common isotope, i.e., more abundant isotope in denominator?

P3L3: Condensable? At what temperature? How about CO? Why not describe the method directly?

P3L17: "types of" is vague. Be more specific or delete it.

P3L24: delete "datasets".

P4L1: "reliable calibrations"? Do you mean being able to reliably characterize the response of your instrument with standards, or do you refer to the standards themselves? The following discussions of "calibrations" is vague.

P4L4: "primary calibrations"? Do you mean calibration of primary standards? What defines the primary standard scale for CH4 isotopes?

P4L9+: What is the Kr interference? I assume it is something with same mass/charge as the "CH4-derived peak"?

P4L11: What is the "CH4-derived peak"?

P4L26: You are summarizing the analytical methods used by each laboratory, not reviewing the labs.

P5L6: In place of standard errors, standard deviations (with "n" given) should be used.

P5L9+: I think the general discussion of techniques (DI-IRMS, GC-IRMS, and optical), calibration, propagation of standards and their traceability to fundamental SI quantities, limitations of current methods, interferences, memory effects (scale compression?), etc. would be useful to isotope data users and fit better here (section 2) rather than in the introduction. This would be a good place to define how metrics like repeatability and reproducibility are quantified and other terms that might be unclear to non-specialists.

P5L15: replace "the early years" with a year or range of years.

P6L19-20: How can calibrations of an instrument in one lab be the basis of calibrations in another lab? Do you mean the standards developed at MPIC were propagated to IMAU?

P6L25: unclear what "calibrations were made against ..." means. What do these abbreviations mean? Was water from these standards injected directly into the spectrometer, or were intermediate standards traceable to them injected? This is used other places, too.

P7L5: what is "a working standard air"?

P7L21: delete "because of"

P9L19-20: Kr interference is significant.

P9L21: Correction of the data for Kr interference ...

P9L23: "Kr removal"? do you mean only these data were corrected for Kr interference?

P9L25: ..RHUL) measured atmospheric ... using....

P10L9: How can one lab share its "calibration" with another. They shared standards?

P10L16: "calibration is made against gas bottles"? What is in the "gas bottles"? What standard is it traceable to?

P10L21-22: "anchored to the INSTAAR calibration"?

P10L23: "overall measurement precision"? How is that different from "precision", repeatability, or reproducibility?

P10L25: What is "an Antarctic bottled air"?

P11L7: You can not transfer a standard scale by measurement of a single sample or even multiple samples. Comparisons of measurements can not replace propagation and maintenance of a standard scale.

P11L18: Dates (or ranges) should be given for each comparison.

P12L6-7: TU filled four ...., two with dry ambient air (give dew point) and two with ... (what is "synthetic standard air"? How is it different from real air?).. with CH4 (delete "concentrations" - the units make it clear what the quantity is (which is not concentration, anyway)).

P12L12: .. after transport to... (throughout)

P12L14-15: "Calibration offsets" is too vague.

P12L21: "scale compression" should be defined.

P12L22: what are the differences in matrix?

P13L12: This title implies something else. Ice cores are not part of the round robin. It is a comparison among labs that measure delta-13CH4 from air extracted from ice cores.

P13L20" "elemental"? Do you mean measurements of XCH4 (i.e., CH4 mole fraction)?

L13L24-26: A change after 9 years could mean a change at PSU, not necessarily drift in the isotopic composition of CH4 in the cylinders. What happened to XCH4? Did it drift? If not, could CH4 isotopic composition change without a change in XCH4? If so, how?

P14L5: The level of agreement depends on your perspective.

P14L14: suite of cylinders...

P15L10: "RHUL-INSTAAR offset"? Are they both different from NIWA, which offsets are calculated from, by the same amount? I suggest "difference" rather than "offset".

P15L12-13: With such large "n", the uncertainty on the difference is deceptively small. SD would be more representative of true uncertainty in the difference, since their could be drifts over time in one measurement vs another.

P15L14: data were corrected...

P15L20: among laboratories

P15l21: delete "It has also happened that"

P15L27-P16L1: Fig. 2 combines....

P16L11: delete "that" or rewrite as "stability ... 1992 to 2007 continues until 2011."

P16L17-18: was the GC-IRMS with the post separation column used to define the empirical correction in L15 or the DI-IRMS?

P16L22,L25: rather than say "high" or "middle" cylinder, why not ...for CH4 at $\sim$X ppb to make it clear?

P17L2: it can not be the "calibration" that is propagated, but rather a standard or standard scale.

P17L3: Since standard scales were never defined, I'm not sure what "primary calibration" refers to? It should be calibration of the instrument at MPI with primary standards, what ever they are.

P17L14: it is the scales, not the calibrations that are important.

P17L14-15: what does "their" refer to? Since Sperlich's affiliation is given as NIWA,

then is this difference between IMAU and NIWA?

P17L22: delete "has".

P18L7: "instrument circumstance" is too vague.

P18L18: both use the same scale.

P18L26: update of the standard scale or the method used to calibrate the response of the instrument with that scale?

P19L12: "intercomparison in this study"? The round robin?

P19L16: ...comparison described by Nisbet?

P20L5: This is the best comparison after correction for Kr interference, so why is excluded from Fig.2a?

P20L18: internal standard?

P21L2: measurements of air in cylinders exchanged between... How many cylinders?

P21L3: applied an offset correction (delete "to").. to all data.. (delete "the")

P22L1: replace "shows the offset to be" with "gives"

P23L10: what makes these "programs"?

P23L11: The results are about measurement offsets; they do not address differences in standard scales directly.

P23L15: among labs...

P24L3: atmospheric CH4 level - when? modern?

P24L26: ...similar to...

---

## Author Comment (AC1) · 9 Dec 2017

We thank the reviewer for his thorough reading and insightful suggestions.

Overall, we very much appreciate the reviewer's comments, in particular from the viewpoint of rigorous technical aspects. We emphasize that, although the issues are not finally solved, the attempt of an estimate of the overall measurement offset as presented in this study will help synthesis and modeling studies, at least as a temporal workaround. This is the original and main scope of this study; however, in the future we will continue efforts from participating laboratories for improving compatibility to maximize use of the available datasets. All of the reviewer's comments are helpful for the steps forward, but here in our response and in revising the manuscript, we focus on the original scope of this study.

Our responses are detailed below, in which **Comments from reviewers** and our responses are given in different styles.
* * *
*Stable isotopes of methane, an important greenhouse gas, are used to understand the budget of methane sources and atmosphere oxidation processes. In particular, isotopes are needed to understand why the growth of atmospheric methane was levelled off in 2006-2007 and later renewed (Nisbet et al., 2016). From 2006 to 2016, δ13C(CH4) is discussed to be shifted to negative direction for ~0.3 ‰ (Nisbet et al., 2016). In order to interpret the isotope data, first of all one has to obtain reliable, high precision and high accuracy data, to be based on reliable calibrations. The problem is that offsets between labs are larger than the measurement reproducibility of each individual lab, in particular the intercomparison Round Robin in 2007–2016 demonstrated discrepancies up to ~0.6 ‰ in δ13C and ~36 ‰ in δ2H (Table 5). In the manuscript, the authors review the situation with calibrations, inter-comparisons and lab-to-lab discrepancies.*

*All in all, the authors have performed a great work to make a summary of numerous measurements done in several labs, their calibration histories and demonstrated discrepancies between data sets produced. However, it is evident that reliable data sets and even data synchronisation cannot be obtained at the*

*moment, and much more work is needed. The authors are in the position to make analysis of the situation and make suggestions on further steps necessary to obtain reliable data in the future; all that has to be listed in the abstract and stressed in conclusions. Suggestions and/or recommendations on calibration practices, use of reference materials and design of inter- comparisons would be valuable both for laboratories and also for Global Atmospheric Watch program at WMO. Biannual WMO-IAEA meetings on CO2 and greenhouse gas measurement techniques are focused on improvements needed, including stable isotope measurements of atmospheric CO2 and methane (e.g. GGMT-2015, GAW Report No 229).*

*The authors express the hope that CH4-mixtures (Sperlich et al., 2016) to be made in the way similar to CO2-in-air calibration mixtures being produced by MPI-BG (JRAS-mixtures) may be of great help. However, GGMT-2015 has recognised that after introducing JRAS-mixtures, lab-to-lab discrepancies in δ13C(air-CO2) demonstrated by intercomparisons have not be decreased (GAW Report No 229) and much work is still needed. Thus, careful analysis of the current situation with methane isotopes as well as focused recommendations need to be give. If the authors are not in a position to come to concise recommendations, at least they should give a better summary of their work and analysis.*

We thank the reviewer for his insightful suggestion. The scope of this study is to collect and present as much information on the currently available data of isotopic composition of atmospheric $CH_4$ as possible in a clear and concise manner. It is true that we need further efforts toward harmonization of the datasets, but we believe that our current best estimate of the inter-laboratory offset is the first important step. A new round robin comparison is being planned for starting next year, and further analysis of the situation and discussion on efforts in individual laboratories should be advanced. In the present study, we focus on a clear description of the current and historical measurements to help following steps to come. We have reformulated the manuscript to improve readability within the scope of this study, but further efforts in the path forward, for instance to

formulate recommendations for measurements is beyond our scope. Such efforts have to be coordinated independently e.g. at WMO-IAEA GGMT meetings and in line with their associated recommendations.

*In general, the manuscript is extremely long, not easy to follow and understand causes of the problems. It can be further optimised by grouping some problems and then addressing these groups. Given that mostly δ13C has been analysed, the reviewer focuses on δ13C data, the same aspects are mostly valid for d2H data.*

*First, the reviewer suggests grouping problems related to calibrations as following (some aspects are addressed below in more details):*

1. *Instrumental effects and raw data corrections. These include (i) cross-contamination (memory) in the mass-spectrometer ion sources; this effect shrinks the δ13C values, namely the distance between NBS19-CO2 used for many calibrations (δ13C of 1.95 ‰) and δ13C values of samples being around -47 ‰; (ii) consistent use of 17O correction for raw CO2 mass-spectrometry data, (iii) Kr-effect in continuous flow mode of mass-spectrometry runs (before optimisations) and its magnitude. Notably, the aspects (i)-(ii) are not addressed in the manuscript.*

We mentioned the cross-contamination effect in section 5 of the original manuscript, but did not do so concerning the $^{17}$O correction issue. We have collected information related to these possible problems from each laboratory and summarized them in an extended table. This table will be added in the revised manuscript. We have also added a new section to present a systematic description on these issues.

2. *Consisted use of Reference Materials (RMs) and data management. Isotope trends can be understood in a reliable way if, and only if data are correctly positioned on the scale. Absence of any drifts can be demonstrated by re-calibrations vs. reliable RMs, to be repeated on regular basis. (The reviewer has not found much information on repeated calibrations.) First, one has to pay attention to the fact that δ13C values of RMs have been revised in the past*

*(e.g. Coplen et al., 2006) and updated values have to be taken. This implies the need for data management & data archiving in the way allowing data reprocessing retroactively. Second, there is no reliable RMs aimed at δ13C in methane, and such dedicated RM(s) is urgently needed. In particular, LSVEC, the high-level RM aimed at δ13C=-46.60 ‰ introduced for data normalisation by (Coplen et al., 2006) is found to be unstable. The scatter in δ13C observed on different LSVEC aliquots is ~0.35 ‰, with the LSVEC value drifting in time due to adsorption of air-CO2 (Assonov et al. in GGMT-2015, GAW Report 229). Calibrations based on NBS19-CO2 & NBS18-CO2 may be biased due to cross-contamination effect (see above) and dedicated instrumental tests have to be performed.*

We agree with the reviewer regarding the importance of RMs and data management. We have added a description of repeated measurements of reference materials where available, and refer to the technical papers from each laboratory for more details. As for the revised $\delta^{13}$C value of RMs, it is true there are laboratories that have not taken into account such corrections. Those laboratories that have ongoing measurement programs over years might consider retroactive data correction, but there are also laboratories that produced a significant amount of data but already closed these activities by now. In these cases, access to raw measurement data and re-evaluation may be laborious and in some cases even impossible. We agree that re-evaluation must become routine for future revisions of RM values. We, the greenhouse gas isotope community, can recommend proper corrections, but the datasets are made available by individual laboratories under their own responsibility. We consider that step-by-step efforts to spread such information broadly is the only way to improve compatibility with updated information for the RMs. We are also concerned for the lack of RMs for $\delta^{13}$C-CH$_4$ and $\delta$D-CH$_4$ and we have therefore extended discussion in the revised manuscript.

3. *Practical calibration approaches and inconsistencies. Careful calibrations and regular re-calibrations is a must. Contrary, a common practice is to transfer calibrations from one lab to another (e.g. by transferring characterised CO2 or CH4 gas, or by performing calibration measurements for another lab, several examples are listed below). This practice cannot be recommended as it may,*

*and often does bring to unpredictable biases (e.g. due to inconsistent use of the 17O correction) and also precludes correct data management (corrections to be applied in a consistent way and/or corrections to be applied retrospectively). This and the fact that revision of δ13C values for RMs took place in 2006 (Coplen et al., 2006), this demands for a careful revision of calibrations in each lab.*

*4. Routine protocols and operating procedures have to be established in each lab and followed, both for calibrations (more critical) and for sample analysis. Any deviation from established protocols (e.g. due to rotation of personnel) may bring to unrecognised bias.*

We thank the reviewer for these comments. We admit that the current situation is undesirable. However, we emphasize that the information that is presented in our paper is the result of time-consuming scientific efforts of individual laboratories. We acknowledge the historical measurements and hope to share the current status broadly, even if severe problems cannot be solved. The problems in scale transfer should be addressed through upcoming comparisons and relevant discussions, together with efforts in each laboratory for a better and accurate link to the VPDB and VSMOW scales. Regarding the routine protocols, an established management has been difficult in each laboratory, because measurement techniques for the isotopic composition of $CH_4$ have been in development over the last few decades. At the same time, even once a certain measurement technique was established, it has been hard for each laboratory to maintain the same measurement circumstance over the years, given that the measurement system is very complex. Therefore, the reviewer's point is still an issue for the path forward.

*In order better summarise all the aspects related to calibrations and make analysis, the reviewer suggests making a large table, listing all the labs and providing columns corresponding to each problem and how it has been addressed. This may be helpful to visualise the situation, adopted calibration approaches and inconsistencies. Without a careful analysis and guidance, numerous examples given in the manuscript may result in some misunderstanding and misleading.*

We appreciate this suggestion. We have prepared an extended table so that readers can better understand the problems and possible causes of the offsets. The table has been attached at the end of this response.

*Second, there is a common misbelief about inter-comparison activities and round robins. These cannot replace factual calibrations (and regular re-calibrations) and also cannot help bringing data on the VPDB δ13C scale in reliable way. The inter-comparisons (e.g. round robin conducted during 2007– 2016 demonstrate large discrepancies, up to 0.6 ‰ in δ13C and 36 ‰ in δ 2H (Table 5) and indicate problems only. As one cannot exclude calibration's biases and/or drifts, non-stable instrumental effects, change in lab 'operation procedures and other problems, lab-to-lab discrepancies in may differ over years. Systematic understanding of inter-comparison results is hardly possible without dedicated design of inter-comparisons; this has to include synchronised use of RMs, synchronised corrections for instrumental effects and synchronised 17O correction. In particular, inter-comparisons have to be done on the same material (same samples or sample archives, or artificial mixtures) and preferentially addressing 1-2 effects (e.g. Kr-related baseline, calibration bias).*

We thank the reviewer for bringing up the important point. As the reviewer says, inter-laboratory comparisons help to harmonize different datasets in a work-around fashion, but pinpointing the merged dataset explicitly on the VPDB/VSMOW scale is different. Accurate anchoring to the standard scale is of course critical in the measurement community to achieve high compatibility. However, our attempt in this study, assessment of the overall measurement offsets, may help data users to some degree since, for instance, a model can be tuned or offset to reproduce a harmonized dataset for a specific scientific topic. This study focuses more on this aspect, current best synchronization of the datasets.

*Third, data management and uncertainty propagation. Similar to air-CO2 isotope data, archiving all raw mass-spectrometry data as well as details of all calibrations can be recommended (e.g. see GAW- 229 Report, GGMT-2015). Then*

*uncertainties can be propagated, in order the uncertainty budget to be used as a tool aimed to demonstrate critical steps and improvements needed. However, the messy situation with calibrations, data management, corrections etc, all that precludes correct uncertainty propagation scheme.*

We agree to the reviewer, and this issue should be addressed in upcoming discussions after we organize the current situation through this study and other opportunities.

*Below some more details are given, starting with the problems related to calibrations:*

*1. Cross-contamination (memory) in the mass-spectrometer ion source is known to shrink the δ13C distance between sample gas and mass-spectrometer reference CO2 gas. The magnitude of cross-contamination effect on MAT252 is reported to be up to a few 0.1 ‰ (Verkouteren et al., 2003a, 2003b). In particular this is relevant to pre-2003 works (e.g. δ13C measurements at MPI-C are done before 2003) when no specific measures have been taken such as Tantalum slits in the ion source and optimisation of ion source tuning.*

*2. All δ13C data are obtained by correcting raw CO2 mass-spectrometry data for 17O contribution, this is so-called the 17O correction. The 17O correction after Craig (1957) modified by Allison et al. (1995) and the one after Santrock et al (1985) has been used for many years; later the 17O correction was re-determined by Assonov & Brenninkmeijer (2003) and this correction has been recommended by IUPAC as the most accurate one and avoiding biases in δ13C (Coplen et al., 2006; Brand et al., 2010). Inconsistent use of the 17O corrections (e.g. by transfer of calibrations from one lab to another) and/or comparing data sets obtained in different years and using different 17O corrections may bring to unpredictable bias(es). Notably, revision of δ13C values for RMs (Coplen et al., 2006) is partly related to the use of 17O-correction after Assonov & Brenninkmeijer (2003).*

*3. One needs to stress that Kr affects the continuous flow mode of mass-spectrometry only; one may give an estimate of the magnitude and its direction.*

We thank for the detailed explanations.

1. We have added texts for cross-contamination effect.

2. We have added texts about the 17O correction, which was not taken into account in the original manuscript.

3. The Kr effect on CF-IRMS measurements was discussed in the introduction, but we have made this more visible by assigning a new subsection. As has been described by Schmitt et al. (2013), the Kr effect varies with time and measurement settings and depends on $CH_4$ mole fraction ($Kr/CH_4$ ratio). In addition, the effect has not been qualitatively identified on a routine basis at some laboratories. Therefore a systematic presentation of magnitude and direction of the Kr effect is difficult.

***The reviewer suggests listing all the cases of calibration' transfer from one lab to another one, again this may be given in a table. Several examples of calibration transfer (not all cases) are listed below:***

- ***from MPIC to IMAU (page 6, lines 5-8);***

- ***from IMAU to MPI-BC (p. 7, l. 5-6), I cite "Initially, the GC-IRMS measurements had been anchored to a working standard air calibrated by IMAU." Note, IMAU has transferred calibration from MPIC, see above.***

- ***from Bundesanstalt für Geowissenschaften und Rohstoffe to MPIC (p. 6, l. 22-23), I cite: "The MPIC δD-CH4 scale is based on measurement of standard gases at the Bundesanstalt für Geowissenschaften und Rohstoffe, Hannover, Germany";***

- ***from UEHI to AWI, I cite: "The δ13C-CH4 measurements follow the UHEI calibration via comparison of measurements of an Antarctic air sample (Möller et al., 2013)."***

- ***from INSTAAR to UB, I cite: "δ13C-CH4 value of −47.34±0.02 ‰ .................. is anchored to the INSTAAR calibration"***

These scale transfer information has been summarized in an extended table prepared in this revision.

***There are examples of incorrect use of δ13C values for RMs.***

- *For UCI' calibration, reference is given to (Rici et al., 2001), and here I cite from Rici : "values of 1.92 and -47.18‰ for NBS-19 and IAEA-CO-9, respectively." There are 2 problems, namely NBS19 has by definition the value of 1.95 ‰ (no revision followed) whereas the latest revision of δ13C values (Coplen at al., 2006) gave for IAEA-CO-9 the value of -47.32 ‰; the difference of 0.14 ‰ is not negligible.*
- *For TU, Umezawa et al. (2009) takes NBS18' recommended δ13C value as 5:029 ‰ whereas the latest revision by Coplen et al. (2006) gives for value of -5.01 ‰ (the difference 0.02 ‰).*

There might be a misreading of Rice et al. (2001). They validated their scale, which had been maintained over years, by measuring $CO_2$ produced from NBS-19 and IAEA-CO-9 against their laboratory standards. The measured values against the UCI scale are those cited by the reviewers. Such validation measurements do not always show complete agreement but this case was in reasonable agreement. It is no wonder that the revised value for IAEA-CO-9 by Coplen et al. (2006) was not used at the time of publication of Rice et al. (2001).

As for the calibration at TU, TU no longer has NBS-18 based $\delta^{13}C$ data open to data users. TU $\delta^{13}C$-$CH_4$ data since Umezawa et al. (2009) are all on the NBS-19 based scale, so there is no inconsistency related to the revision of NBS-18 value.

*There may be cases of unexplained drifts and effects of large magnitude, such as one reported for TU. The manuscript refers to Umezawa et al. (2012) and, I cite from this work: "The measured 13C value of the test gas was stable to be (−47.12±0.10) ‰ from the start date of our measurement until April 2008. Afterward, the measured value suddenly shifted to (−46.85±0.09) ‰, keeping the same precision as before. The cause is still unclear, since we had not changed any measurement settings. To keep data consistency, we added −0.27‰ to the measured values after the gap."*

Such shifts can happen in reality. Some laboratories have exchanged samples, co-operated co-located samplings and so on to help find these drifting issues. Relevant information has been given in the revised table.

---

## Author Comment (AC2) · 9 Dec 2017

We thank the reviewer for the thorough reading for the many corrections and the insightful suggestions, especially for the rigor in metrology and terminology and suggestions to make this paper more broadly acceptable in the $CH_4$ research community.

According to the reviewer's suggestion, in the revised manuscript, we have added a new section that addresses an overview of the IRMS-based measurement techniques for isotope ratios of $CH_4$, standard scales and current problems that may contribute to the observed measurement offsets among laboratories. Accordingly, related places in other sections have been also reformulated. We hope this additional introduction will help non-expert readers to follow the contents.

We have defined use of the term "calibration" in the revised manuscript. It means in this study a measurement of a gas at a laboratory against a standard at higher hierarchy level of the standard scale and to assign the gas a $\delta^{13}C$-$CH_4$ or $\delta D$-$CH_4$ value traceable to the standard scale. According to this, we found the term "calibration offset" used in the original manuscript inaccurate, because the offsets summarized here do not necessarily come from calibration only (as pointed out by the other referee Dr. Sergey Assonov). We use "measurement offset" as suggested by the reviewer.

Our responses are detailed below, in which **Comments from reviewers** and our responses are given in different styles.
* * *
***The authors take on the difficult and important task of summarizing comparisons of measurements of CH4 stable isotopes in air at CH4 levels appropriate for the current atmosphere and air extracted from ice cores. Done properly, this would give data users correction factors to combine data from different laboratories and give them an understanding of the major issues involved so they fully understand the limitations of the combined data sets. This would allow more CH4 isotope data to be used in studies of the global CH4 budget. Unfortunately, the paper seems to be written for isotope measurement experts, like the authors, rather than for data users who may be very interested in more CH4 isotopic***

*composition measurements in their studies. It contains too much jargon and too little explanation of the major issues that prevent labs from preparing a combined data set for CH4 isotopes with meaningful temporal and spatial gradients. These issues seem to be inherent in the community's measurement approach. But what are those issues? Are they associated with deficiencies in measurement techniques themselves and how instruments are calibrated? Is there a lack of appropriate isotope standards for CH4? Is the issue with propagating standard scales from carbonates (13C) or water (D/H) to CH4 in air? The fundamental hierarchy of standards used within the CH4 isotope community should be described; much of what is described seems to violate good metrological practice. Are sample collection or processing methods causing differing amounts of fractionation among labs? These are just some possibilities from a non-expert. I suggest that a brief, systematic description of the important issues involved in making measurements of atmospheric CH4 isotopic composition and ultimately preparing a consistent data set across measurement labs is given in Section 2. This would be especially useful to data users and also help the authors focus on how to move forward.*

We thank the reviewer for the valuable comment. According to the reviewer's suggestion, we have added a new section to present systematic descriptions of the measurement overview and critical issues. We have modified the manuscript at related places accordingly.

*This study is supposed to help scientists utilize more CH4 isotopic data by providing correction factors to make data from different labs more compatible, but are the data sets available? A quick look at the World Data Center shows only NIWA and NOAA data have been updated within the past couple years for 13CH4, and the only other data set, quite outdated, is from Tohoku University. Are the data from ice core and firn air available? Do other labs make data available through their institutes' web sites? That issue aside, using the comparison information in the paper is complicated by Kr interference; it is not always clear where a correction has been applied and, as a result, what data sets the information in this paper is appropriate for. It gives the impression that the paper*

*was written before the isotope measurement community is ready for such an effort to be more generally useful to the CH4 research community.*

We have compiled information on data availability from each laboratory and it is included in the revised manuscript. We have also clarified the procedures and corrections for the Kr interference. We admit that not all problems are resolved by our manuscript. To reach this in the future, we need to establish a forum to advance compatibility of the datasets. This study addresses the current reality and establishes a baseline for steps forward. We believe that our study will help future efforts in the isotope measurement and also broader $CH_4$ research communities.

*It was surprising that XCH4 (i.e., CH4 mole fractions) was not reported in the paper, at least for the cylinders that were circulated as part of this study. It is important that CH4 measurements agree among labs measuring air from ice cores and the modern atmosphere to get radiative forcing correct, and this is a good forum to show that level of agreement.*

Intercomparison of measurements of $CH_4$ mole fraction is out of scope of this study. Laboratories participating in this study are all specialized for isotope measurements, but we do not investigate mole fraction measurements in each laboratory. There are established activities rigorously concerning on compatibility issues of $CH_4$ mole fraction (regular GGMT meetings and following WMO/GAW report).

*The manuscript is poorly written. It is far too wordy. While the first author is not a native English speaker, there are at least 10 authors who are. Shame on all of you for not improving the English. Language is often vague. Scientific terms are misused.*
*Overall, I think the work described in the paper is important, despite that the community is not yet capable of preparing useful combined data sets. I recommend that the paper is re-evaluated for acceptance after the authors respond to the comments in this review to the satisfaction of the journal editor.*

We apologize and we are deeply ashamed. We have revised the text to improve

conciseness and formulations.

*General comments: 1. Use appropriate metrological terms. "Precision" is a qualitative term, yet it is used as a quantitative measure of the noise or uncertainty in a measurement system. Is it short-term noise (repeatability) or does it represent long-term variations of a measurement system (reproducibility)? When the proper terms are used, how are they quantified? More appropriately, uncertainty should be stated with its confidence interval.*

We have revised and complemented the text. Concerning the measurements at many laboratories, however, detailed descriptions about how to evaluate the quality of the respective measurements have been given in the literature. We avoid repetition and refer to the original papers from each laboratory.

*2. Calibration: paraphrased, it links the measured response of an analyzer to the known values of standards. In the text, it seems to be confused with standard, and its use is often unclear. Given that, terms like "calibration offsets" are vague. Are the standards different? Is the issue with propagation of the standards? Could the offset result from fractionation during sample processing?*

We thank the reviewer for this comment. We have reformulated the texts to avoid confusions. We have defined the term "calibration" and the term "standard" is now used more rigorously in the revised manuscript.

*3. Differences between labs are often given with standard errors; I think standard deviation would be a better metric. In cases where n is large, e.g., for ongoing comparisons that happen over years, standard error exaggerates how well the difference is known.*

We partly agree to the reviewer. We have replaced standard errors with standard deviation at some places where large number of n could be misleading, but in this study we are interested in difference of the mean of measurements expected for the whole data population in individual laboratories (represented by standard error) rather than

difference of measurements with limited number from one-shot comparison (represented by standard deviation).

**4. Remove unnecessary words: assessed to be... comparison exercises (delete "exercises") evaluated to be X L in volume (delete "in volume") considered to be "in the time period" and "the years" offset value (delete "value")**

We have revised the text for conciseness throughout the manuscript.

**5. Each participant in measurement of CH4 isotopes from the circulated cylinder should report XCH4.**

As written for the earlier comment, comparison of $CH_4$ mole fraction measurements is beyond the scope of this paper.

**6. "Concentration" is misused. In most cases, it can be deleted, because the unit provided (ppb) defines the measured quantity.**

We used the term "concentration", because it has been used conventionally over long years in the greenhouse gas research community. In the revised manuscript, we have replaced the term with "mole fraction".

**7. Why were other labs measuring CH4 isotopic composition not included (e.g., Oregon State University)?**

We believe that we have covered all laboratories that specialize isotope measurements of atmospheric $CH_4$ at operational level. By personal communication, we have confirmed that the group studying air from ice core samples at Oregon State University does not measure $CH_4$ isotope ratios.

**8. What are the main sources of differences among labs? You imply it is differences in standard scales, but why? Don't the working standard scales propagate back to a primary scale, e.g., VPDB for 13C?**

This is related to the earlier general comment 2. We seem to have confused the reviewer due to inconsistent use of terms like "scale" and "calibration". Possible causes are multiple such as differences in instrument settings (including the Kr interference), use of reference materials and resultant realization of the standard scale, and data correction and management applied in each laboratory. For clarity, all laboratories report on the VPDB/VSMOW scale. According to the reviewer's suggestion, we have added a section to describe related information more systematically.

**9. What is the path forward, beyond what was mentioned regarding newly-developed standards? Many deviations from good metrological practice, especially regarding propagation of isotope standard scales, have occurred within this community. The new standards, although developed using an approach that may not defensible in a pure metrological sense, seems practical given the limited resources of the measurement community. Is that alone sufficient? What else needs to be done to make existing data more compatible? How could new laser-based spectroscopic methods help this measurement community and the science? What else could improve the quality and compatibility of measurements of CH4 isotopic composition across the labs involved here and beyond to others? As mentioned, data availability is not considered.**

We thank the reviewer for this comment. It is true that metrological practice has not been complete within the $CH_4$ isotope measurement community; individual laboratories have made efforts for best scientific outcome with limited resources at that time. We hope that our publication will lead to a discussion on the path forward for improving compatibility of available datasets. This study, which summarizes historical and currently ongoing measurements, is the first step. A new round robin comparison for isotope ratios of $CH_4$ is already planned as a next step, which is the first attempt for direct comparison among most laboratories that have currently ongoing measurement programs (in contrast our paper reports an estimate of offsets based on a number of patchy comparisons in history). In the meantime we need to establish a discussion forum suitably for instance in the GGMT meeting to better address causes of

measurement offsets and unify data management in different laboratories in order to decrease the offset. We can make these efforts together with laboratories operating ongoing programs. For existing datasets from laboratories that have closed down their isotope measurement programs, offsets given in this study represent the best estimates and a significant change is not be expected unless new information is brought up through upcoming discussions.

Progress in optical techniques is impressive. For instance measurement of mole fractions has become less cumbersome. New laser-based measurements will likely help to improve isotope analysis, avoiding the need for chemical conversions which could cause specific isotopic fractionations. Since an IRMS-based measurement is still a standard method, careful studies to establish the relationship between IRMS- and laser-based techniques are needed. Discussion on usability of laser spectroscopy is however beyond the scope of this study.

Regarding data availability, we have provided a table that includes how to access datasets.

**Specific comments:**

**P1L32: suggest ..from an inter-laboratory comparison of measurements.... (Also for title.)**

We have corrected the sentence as suggested. The manuscript title has been changed to: Inter-laboratory comparison of $\delta^{13}C$ and $\delta D$ measurements of atmospheric $CH_4$ for combined use of datasets from different laboratories.

**P1L32: ..among worldwide...**

Corrected as suggested.

**P2L3: What does "the data" refer to? The differences among labs?**

We have changed the sentence to: "the difference among laboratories at modern atmospheric $CH_4$ level spread…"

***P2L4-5: As presented, it is not clear how this will help combine data sets. It could be more clear it a table was given of available data sets and if (i.e., with respect to Kr interference), and how, the offsets in the paper apply.***

We will present a table listing available datasets and describe how to apply the offsets in the discussion section.

***P2L8-12: The description of how CH4 isotopes are used to constrain the CH4 budget could be stated more clearly. I suggest something like "The mass-weighted average delta-13C of emissions from all sources will equal the delta-13C of atmospheric CH4, after correction for fraction by removal processes." While you give some references for studies that use isotopes of CH4 (some are poor examples), you don't reference early literature that identified their usefulness (e.g., Stevens).***

We have added the following sentence and a citation of Stevens and Rust (1982) and Cicerone and Oremland (1988) here.

Dictated by global mass balance, the average isotopic composition of $CH_4$ in the atmosphere ($\delta^{13}C$-$CH_4$ or $\delta D$-$CH_4$) equals the flux-weighted isotopic composition of the sources, corrected for the total kinetic isotope effects of removal processes (e.g. Stevens and Rust, 1982; Cicerone and Oremland, 1988; Quay et al., 1991, 1999; Miller et al., 2002; Turner et al., 2017; Rigby et al., 2017).

***P2L21: Is not this ratio more generally rare/common isotope, i.e., more abundant isotope in denominator?***

We thank the reviewer for this correction. The sentence has been changed to:
"…R represents the atomic ratio of the less abundant over the most abundant isotope in the sample and the standard, respectively."

***P3L3: Condensable? At what temperature? How about CO? Why not describe the method directly?***

Since a detailed technical explanation is out of scope, we keep a brief description, but changed the sentence to:

The original methodology was based on the combustion of $CH_4$ in sample air, but interfering compounds such as $CO_2$, $H_2O$, $N_2O$, CO and nonmethane hydrocarbons had been removed cryogenically, chemically or by gas chromatography before combustion of $CH_4$.

**P3L17: "types of" is vague. Be more specific or delete it.**

We have changed the sentence to: …to quantitatively separate different $CH_4$ source categories (…).

**P3L24: delete "datasets".**

Corrected as suggested.

**P4L1: "reliable calibrations"? Do you mean being able to reliably characterize the response of your instrument with standards, or do you refer to the standards themselves? The following discussions of "calibrations" is vague.**

Here we meant accuracy of measurements, that is, reliable link to the standard. The original wording was replaced to "traceability to the standard scale". We have defined the term "calibration" in the revised manuscript.

**P4L4: "primary calibrations"? Do you mean calibration of primary standards? What defines the primary standard scale for CH4 isotopes?**

The standard scale for $\delta^{13}C$-$CH_4$ ($\delta D$-$CH_4$) is VPDB (VSMOW), and here we use the term calibration as a measurement of laboratory reference gases relative to certified reference materials to link ambient air measurements in the laboratory to the standard scale. We have deleted the term "primary".

**P4L9+: What is the Kr interference? I assume it is something with same**

*mass/charge as the "CH4-derived peak"?*

We have added the following sentences:
Schmitt et al. (2013) demonstrated that the doubly charged krypton isotope $^{86}Kr^{2+}$, produced in the ion source of an IRMS, can cause lateral tailing extending into the Faraday cups used for $\delta^{13}C$ analysis (i.e. m/z of 44, 45 and 46), which compromises the measured signal of the $CH_4$-derived peak.

*P4L11: What is the "CH4-derived peak"?*

We have changed the sentence to: …from the $CO_2$ peak generated from $CH_4$ oxidation in sample air (hereafter $CH_4$-derived peak) (Schmitt et al., 2013).

*P4L26: You are summarizing the analytical methods used by each laboratory, not reviewing the labs.*

We have changed the word "review" to "summarize" in the sentence.

*P5L6: In place of standard errors, standard deviations (with "n" given) should be used.*

See our response to the reviewer's general comment 3.

*P5L9+: I think the general discussion of techniques (DI-IRMS, GC-IRMS, and optical), calibration, propagation of standards and their traceability to fundamental SI quantities, limitations of current methods, interferences, memory effects (scale compression?), etc. would be useful to isotope data users and fit better here (section 2) rather than in the introduction. This would be a good place to define how metrics like repeatability and reproducibility are quantified and other terms that might be unclear to non-specialists.*

According to this suggestion, we have added a new section and reformulated the related sections.

***P5L15: replace "the early years" with a year or range of years.***

We have replaced it with the year 1988.

***P6L19-20: How can calibrations of an instrument in one lab be the basis of calibrations in another lab? Do you mean the standards developed at MPIC were propagated to IMAU?***

It is indeed propagation of the standard scale from one laboratory to another. We have tidied up the confusing use of "calibration".

***P6L25: unclear what "calibrations were made against ..." means. What do these abbreviations mean? Was water from these standards injected directly into the spectrometer, or were intermediate standards traceable to them injected? This is used other places, too.***

For clarity, we have added an explanation about these standards at an early part of this section.

***P7L5: what is "a working standard air"?***

It means air (filled in a cylinder) that was measured against standards at higher hierarchy level and that is routinely used to link sample measurements to the standard scale. We have given a description of use of these terms in the newly added section.

***P7L21: delete "because of"***

Corrected as suggested.

***P9L19-20: Kr interference is significant.***

Corrected as suggested.

**P9L21: Correction of the data for Kr interference ...**

Corrected as suggested.

**P9L23: "Kr removal"? do you mean only these data were corrected for Kr interference?**

We have changed the sentence to: …have not been interfered by Kr.

**P9L25: ..RHUL) measured atmospheric ... using....**

Corrected as suggested.

**P10L9: How can one lab share its "calibration" with another. They shared standards?**

The term "calibration" was replaced with "standard scale".

**P10L16: "calibration is made against gas bottles"? What is in the "gas bottles"? What standard is it traceable to?**

The gas bottles are filled with $H_2$ gas measured and certified by the Oztech Gas Company. We have added "$H_2$" in the sentence.

**P10L21-22: "anchored to the INSTAAR calibration"?**

The sentence was changed to: This standard value is anchored to the standard scale used at INSTAAR (…).

**P10L23: "overall measurement precision"? How is that different from "precision", repeatability, or reproducibility?**

Measurement from ice core sample needs extraction of air occluded in ice. This is an additional procedure compared to measurements of sample air made at other laboratories. We have changed the sentence to: The overall measurement reproducibility for ice core sample (including extraction of air from an ice sample) was …

**P10L25: What is "an Antarctic bottled air"?**

It was replaced by "a high pressure cylinder filled at the Alert Station".

**P11L7: You can not transfer a standard scale by measurement of a single sample or even multiple samples. Comparisons of measurements can not replace propagation and maintenance of a standard scale.**

The sentence was modified to: The $\delta^{13}$C-CH$_4$ measurements are linked to the UHEI standard scale via comparison of measurements of an Antarctic air sample (…).

**P11L18: Dates (or ranges) should be given for each comparison.**

Measurement dates or periods are given wherever known.

**P12L6-7: TU filled four ...., two with dry ambient air (give dew point) and two with ... (what is "synthetic standard air"? How is it different from real air?).. with CH4 (delete "concentrations" - the units make it clear what the quantity is (which is not concentration, anyway)).**

We have changed the term to "synthetic CH$_4$-in-air gas". This type of gas is produced by diluting pure CH$_4$ gas with synthetic air, which is a mixture of N$_2$ and O$_2$ at atmospheric fractions (for some cases plus Ar). We have changed the term "concentration" to "mole fraction".

**P12L12: .. after transport to... (throughout)**

Corrected as suggested.

*P12L14-15: "Calibration offsets" is too vague.*

We replace the term with "difference".

*P12L21: "scale compression" should be defined.*

We have added an explanation in the revised manuscript.

*P12L22: what are the differences in matrix?*

The ambient air (MD1 and MD2) was made by compressing natural ambient air with only water vapor removed. It consists of major atmospheric components ($N_2$, $O_2$ and Ar) with many atmospheric trace gases ($CO_2$, $CH_4$, hydrocarbons etc.). In contrast, the synthetic $CH_4$-in-air gas was produced by diluting pure $CH_4$ gas with so-called synthetic air from which most carbonated compounds were removed. Presence/absence of specific gas and resultant difference in mole fraction can potentially affect individual processes of $CH_4$ isotope measurements. We have changed the sentence as follows: …difference in air matrix i.e. natural versus synthetic air (potential influence of the composition of the bulk gas)…

*P13L12: This title implies something else. Ice cores are not part of the round robin. It is a comparison among labs that measure delta-13CH4 from air extracted from ice cores.*

The subsection title was changed to: Round Robin Comparison among Ice Core Analysis Laboratories.

*P13L20" "elemental"? Do you mean measurements of XCH4 (i.e., CH4 mole fraction)?*

Since we present the isotopic composition of $CH_4$ only in this study, the term was deleted.

***L13L24-26: A change after 9 years could mean a change at PSU, not necessarily drift in the isotopic composition of CH4 in the cylinders. What happened to XCH4? Did it drift? If not, could CH4 isotopic composition change without a change in XCH4? If so, how?***

We cannot exclude possibility that any instrument condition at PSU changed after 9 years, but the measurements for two of the three cylinders were in agreement before and after the 9 years, indicating that a change at PSU is less likely.

Concerning change in $CH_4$ mole fraction, measurements before and after the round robin indicate slight drifts (1–2 ppb) for the cylinder with the middle and low $CH_4$ mole fractions (CA 71560 and CA 01179, respectively). As the reviewer questions, the change in $\delta^{13}C\text{-}CH_4$ may be associated with the drift in $CH_4$ mole fraction. However, the determinate cause, including relation between $\delta^{13}C\text{-}CH_4$ and $CH_4$ mole fraction, has yet to be resolved.

***P14L5: The level of agreement depends on your perspective.***

Indeed, however, here we give the number for level of agreement (0.37 ‰).

***P14L14: suite of cylinders...***

Corrected as suggested.

***P15L10: "RHUL-INSTAAR offset"? Are they both different from NIWA, which offsets are calculated from, by the same amount? I suggest "difference" rather than "offset".***

Corrected as suggested.

***P15L12-13: With such large "n", the uncertainty on the difference is deceptively small. SD would be more representative of true uncertainty in the difference, since their could be drifts over time in one measurement vs another.***

According to the suggestion, we use SD for these differences.

**P15L14: data were corrected...**

Corrected as suggested.

**P15L20: among laboratories**

Corrected as suggested.

**P15l21: delete "It has also happened that"**

Corrected as suggested.

**P15L27-P16L1: Fig. 2 combines....**

Corrected as suggested.

**P16L11: delete "that" or rewrite as "stability ... 1992 to 2007 continues until 2011."**

Corrected as suggested.

**P16L17-18: was the GC-IRMS with the post separation column used to define the empirical correction in L15 or the DI-IRMS?**

The sentence was misleading. We have changed it to: The GC-IRMS system is currently equipped with a post-combustion separation column to eliminate the Kr interference.

**P16L22,L25: rather than say "high" or "middle" cylinder, why not ...for CH4 at ~X ppb to make it clear?**

We have specified the mole fractions or cylinder numbers at various places.

**P17L2: it can not be the "calibration" that is propagated, but rather a standard or standard scale.**

The term has been replaced with "standard scale".

**P17L3: Since standard scales were never defined, I'm not sure what "primary calibration" refers to? It should be calibration of the instrument at MPI with primary standards, what ever they are.**

We have defined the standard scale and the term "calibration" in the revised manuscript.

**P17L14: it is the scales, not the calibrations that are important.**

Corrected as suggested.

**P17L14-15: what does "their" refer to? Since Sperlich's affiliation is given as NIWA, then is this difference between IMAU and NIWA?**

The sentence has been corrected to make it clear.

**P17L22: delete "has".**

Corrected as suggested.

**P18L7: "instrument circumstance" is too vague.**

The sub-sentence has been deleted.

**P18L18: both use the same scale.**

Corrected as suggested.

**P18L26: update of the standard scale or the method used to calibrate the response of the instrument with that scale?**

"Calibration" has been replaced with "standard scale".

**P19L12: "intercomparison in this study"? The round robin?**

This has been corrected to: The intercomparison between UHEI and MPI-BGC in this study…

**P19L16: ...comparison described by Nisbet?**

This has been changed to: …an intercomparison presented by Nisbet (2005), …

**P20L5: This is the best comparison after correction for Kr interference, so why is excluded from Fig.2a?**

The Kr interference was not removed for this comparison. We have nevertheless added this result to the figure.

**P20L18: internal standard?**

We keep this term as is, because it does not necessarily mean drift of the standard scales but possibly erroneous propagation of standard scales to internal working gases (which we refer to as "calibration" in this study) at either laboratory.

**P21L2: measurements of air in cylinders exchanged between... How many cylinders?**

Corrected as suggested. We have also added the number of cylinders (2).

**P21L3: applied an offset correction (delete "to").. to all data.. (delete "the")**

Corrected as suggested.

**P22L1: replace "shows the offset to be" with "gives"**

Corrected as suggested.

**P23L10: what makes these "programs"?**

We have replaced "programs" to "results".

**P23L11: The results are about measurement offsets; they do not address differences in standard scales directly.**

We have replaced "calibration" to "measurement".

**P23L15: among labs...**

Corrected as suggested.

**P24L3: atmospheric CH4 level - when? modern?**

We have added the word "modern".

**P24L26: ...similar to...**

Corrected as suggested.

---

## Author Response (AR1)

[revised manuscript text omitted]

---

## Author Response (AR2)

Dear Professor Keppler,

Thank you for handling our manuscript titled "Inter-laboratory comparison of $\delta^{13}C$ and $\delta D$ measurements of atmospheric $CH_4$ for combined use of datasets from different laboratories". We appreciate your comments to make our manuscript better consistent. We have revised our manuscript and our responses are described below.

***Editor's comment***
—Our response

***Dear Taku,***
***your manuscript has substantially improved by taking into account the comments of the 2 reviewers. However, before the manuscript can be finally accepted for publication there are some minor issues which need to be considered.***
***- In Table 2 most of the uncertainties provided in column 3 and 6 (isotope values measured by UCI) are based on one or two measurements (N=1 or N=2). How is this possible? Please clarify?***
—It was our mistake that we had not corrected uncertainties so as to be consistent throughout the paper. Now we do not present any uncertainty for the $N$=1 measurements and we give differences of $N$=2 measurements in parenthesis as in Table 4. We have added explanations in the footnote of the table.

***- In Table 4, the uncertainties provided in column 4 and 7 (isotope values measured by UHEI) and column 8 (isotope values measured by MPI-BGC) UCI) are based on N=1 or N=2. Please clarify?***
—The original table included typical reproducibility estimated from measurements of a standard gas or a quality control sample at UHEI. To keep consistency throughout the manuscript, we have left out the uncertainty numbers for $N$=1 measurements as in the other tables. There were also our mistake by which single measurements ($N$=1) were written wrongly (as $N$=2), which has been now corrected. To include the original intention to indicate the reproducibility when the measurements were made, we have added the following sentence in section 4.3.

"The typical reproducibility for the measurements is between 0.02 and 0.05 ‰ for $\delta^{13}$C-CH$_4$ and between 1.6 and 2.6 ‰ for $\delta$D-CH$_4$."

*- Furthermore in Tables 2 to 5 you should make clear (below the Table) what the provided uncertainty range means (standard deviation or standard error). Alternatively you could also refer to the original papers.*

—According to the suggestion, we have added explanations in the footnotes of each table.

*Finally, I very much would like to thank the 2 reviewers for their efforts that led to a considerable improvement of the manuscript.*

*Best regards*

*Frank*

We all are also grateful to the reviewers and the editor for enabling us to greatly improve this manuscript.

Best regards,

Taku Umezawa

[revised manuscript text omitted]